# *Toxoplasma gondii* infection induces early host cell cycle arrest and DNA damage in primary human host cells by a MYR1-dependent mechanism

Zahady D. Velásquez ✉, Lisbeth Rojas-Baron, Iván Conejeros, Carlos Hermosilla & Anja Taubert

*Toxoplasma gondii*, an obligate intracellular parasite, control its host cell cycle through mechanisms that are not fully understood. Key effector molecules, including MYR1 and HCE1, play roles in translocating parasite proteins and inducing host cellular cyclin E1 overexpression, respectively. We investigated the early role of MYR1- and HCE1-driven host cell cycle arrest and DNA damage (up to 3 h p.i.). Our findings showed that *T. gondii*-infected cells experienced S-phase arrest and displayed double-strand DNA breaks as soon as 15 min p.i. This condition persisted until 3 h p.i., at which point we also observed increased host cell binucleation and micronuclei formation, both hallmarks of genomic instability. Furthermore, host cells responded to DNA damage by activating the ATM branch of the homologous recombination repair pathway. MYR1 was shown to be crucial, as TgΔmyr1 tachyzoites failed to induce S-phase arrest and DNA damage foci. In contrast, the absence of HCE1 did not produce these effects, suggesting that cyclin E1 expression was not involved. Also, DNA damage was demonstrated to be ROS-independent, suggesting that ROS did not trigger DNA damage. Our results suggest that *T. gondii* compromises host cellular DNA integrity depending on MYR1 shortly after infection, maintaining it over time.

T*oxoplasma gondii* is an obligate intracellular apicomplexan parasite and the aetiological agent of toxoplasmosis. As a polyxenous parasite, *T. gondii* infects almost any homeothermic animal, including approximately one-third of the human population worldwide[1,2]. In intermediate hosts (IH), tachyzoites disseminate and differentiate into cyst-forming bradyzoites, which are transmitted to definitive hosts (DH), i.e., members of the Felidae family[3]. For successful intracellular replication, *T. gondii* tachyzoites invade nucleated host cells with the help of several proteins being mainly released from the parasite´s rhoptries and micronemes, both representing specialised secretory organelles found within the Apicomplexan subphylum (phylum Alveolata). In addition, dense granules are vesicular organelles containing GRA proteins, which represent key excretory/secretory proteins of *T. gondii*. GRA proteins play a pivotal role in facilitating parasite intracellular establishment and they either remain within the parasitophorous vacuole (PV) or are translocated into the host cell cytosol or nucleus[4]. Overall, parasite excretory/secretory products are involved in the acquisition of nutrients, lipids and soluble proteins, host organelle recruitment and cell cycle progression control, amongst others[5–9]. Referring to the control of host cell cycle progression, *T. gondii* tachyzoite stages of different haplotypes (e.g., I, II, II) were reported to arrest host cells in S- or G2/M-phase, thereby preventing host cells from proper cell division[10–14]. In this context, *T. gondii*-derived effector molecules like MYR1 (c-*myc* regulation 1) and HCE1 (host cyclin E) upregulated the expression of host cellular cyclin E1, an important cell cycle regulator[7,15]. MYR1 is located at the PV membrane forming part of a multiprotein complex, which mediates the translocation of several parasite-derived effectors like GRA16 or GRA24 across the PV membrane into host cytosol and nucleus, thereby controlling host cellular *c-myc* activity[5,15,16]. Thus, the MYR1-dependent dense granule effector GRA16 was reported to be involved in p53-related and cell cycle pathways[5]. c-Myc is a key transcription factor that regulates several critical host cell processes, such as cell cycle progression, cell metabolism, and apoptosis[17]. In consequence, host cells infected with a MYR1-depleted *T. gondii* strain are unable to control host cell cycle progression[15]. In dependence on MYR1, HCE1 induces cyclin E1 overexpression in host cells, therefore, HCE1

Institute of Parasitology, Biomedical Research Center Seltersberg, Justus Liebig University Giessen, Giessen, Germany.
✉e-mail: zahady.velasquez@vetmed.uni-giessen.de

depletion hampers host cellular cyclin E1 upregulation and leads to a loss of host cell cycle progression control by the parasite[7]. Interestingly, *T. gondii* also affects mitosis in host cells by inducing both the formation of aberrant mitotic spindles with multiple centrosomes and centromeres and cytokinesis failure[14,18]. Overall, these processes have the potential to affect host cell genome integrity. Therefore, we here aimed to analyze whether *T. gondii* induces DNA damage in primary human endothelial host cells and whether this finding is related to specific parasite-derived molecules and cell cycle progression control. To tackle the onset of these parasite-mediated effects and to avoid overlapping effects driven by parallel parasite mitosis, we here focused on the first three hours of infection. To avoid artefacts resulting from DNA damage events driven by host cell immortalisation or tumour origin, we here worked with primary human umbilical vein endothelial cells (HUVEC). Current data show that infections with *T. gondii wt* and HCE1-deficient tachyzoites result in host cellular DNA double-strand breaks as early as 15 min p. i. This effect proved absent in infections with a MYR1-deficient strain, thereby suggesting that DNA damage occurs in a cyclin E1-independent but MYR1-dependent fashion. We furthermore show that *T. gondii* induces host cell binucleation, micronuclei formation and aberrant mitosis very soon after infection (3 h p. i.) and independent of parasite HCE1- and MYR1 expression.

## Results

### *Toxoplasma gondii* arrests the host cell cycle early after infection in a MYR1-dependent mechanism

In the first experimental series, we intended to analyse *T. gondii* infection-mediated host cell cycle alteration early after infection (within the first 3 h of infection) to avoid tachyzoite replication-driven artefacts and to identify the

onset of host cell cycle impairment. Moreover, we aimed to analyse the role of the parasite effector molecules MYR1 and HCE1 in cell cycle modulation by using respective *T. gondii* knockout strains (*Tg*Δmyr1 and *Tg*Δhce1 mutants, respectively). Therefore, primary human umbilical vein endothelial cells (HUVEC) were infected with *Tg*Δmyr1, *Tg*Δhce1 and control strain (*Tg*Δku80) tachyzoites at a low ratio (0.5:1) and fixed at 15 min (=0.25 h), 1 h and 3 h p. i. HUVEC were then analysed for the nuclear presence of two classical cell cycle markers, geminin (G2-phase) and PCNA (S-phase). In this assay, the absence of both markers (geminin and PCNA) indicates cells in the G1-phase (for illustration, see Fig. 1A).

The current data revealed that *T. gondii wt* infections led to a rapid arrest of host cells in the S-phase. Thus, already after 15 min of infection, a significantly increased proportion (35.4%, $p = 0.0003$) of cells within the total HUVEC layer showed S-phase characteristics in comparison to non-infected cell layers (9.9%) and remained in this cell cycle phase until the end of the experiment at 3 h p. i. (Fig. 1B). Given that we used a low MOI, not every single cell within an infected cell layer was infected with *T. gondii*. However, $63.6 \pm 0.02\%$ of cells arrested in the S-phase revealed indeed to be infected with *T. gondii* tachyzoites (Supplementary Fig. S1A). So far, it remains unclear whether *T. gondii* infection-mediated defects in host cell cycle progression are driven directly by the parasite or whether they represent collateral effects of the infection or parasite development. It is well known that *T. gondii* actively manipulates its host cell via parasite-secreted proteins that directly interact with the molecular host cell machinery. The secreted *T. gondii* MYR1 protein helps in the parasite-derived HCE1 translocation into the host cell nucleus thereby regulating the activity of the transcription factor E2F/DP[15,16,19,20]. Consequently, RHΔmyr1 and RHΔhce1 mutants resulted in a lack of host cell cyclin E1 upregulation[7,15].

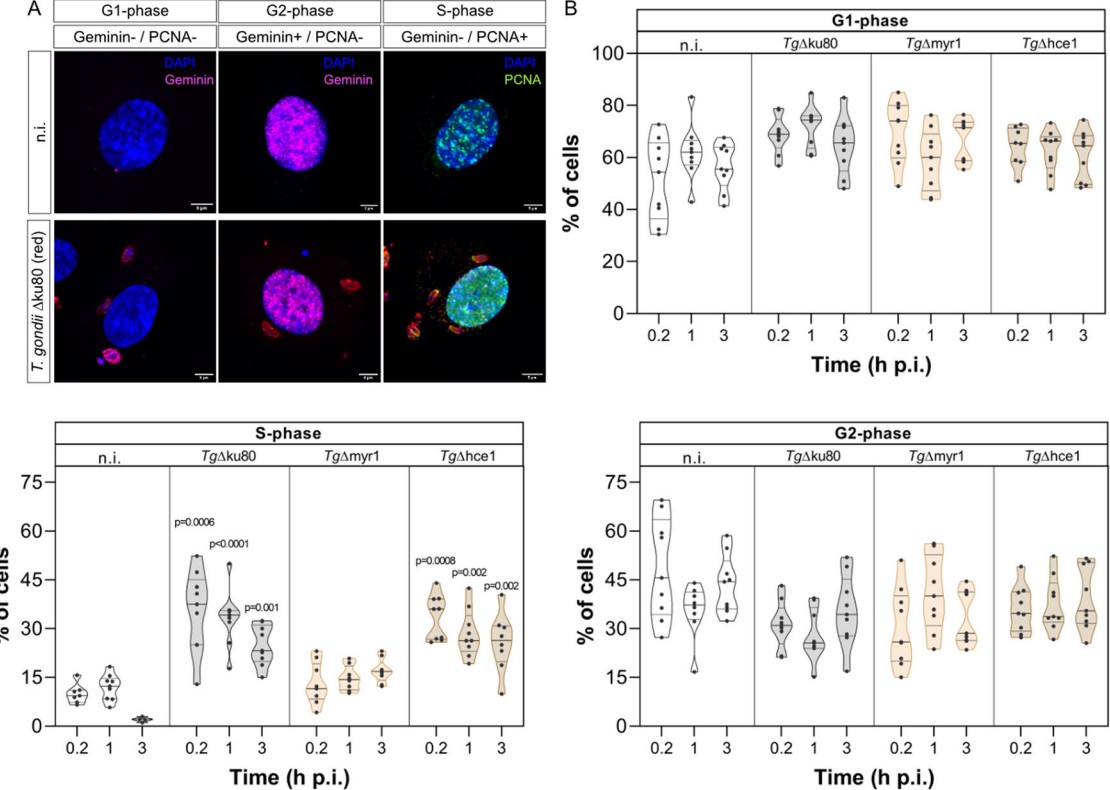

**Fig. 1 | *T. gondii* infection induces S-phase arrest as early as 15 min p.i. A, B** HUVEC (*n* = 3) were infected with *T. gondii* tachyzoites (MOI 0.5:1) of *Tg*Δhce1, *Tg*Δmyr1 mutants and *wt* parasites (*Tg*Δku80) and fixed at 15 min, 1 h and 3 h p. i. Cells were stained in parallel for geminin (G2-phase marker, magenta), PCNA (S-phase marker, green) and DAPI (DNA maker, blue) (**B**). The total number of cells in G2-phase (geminin-positive/PCNA negative), S-phase (PCNA-positive/geminin negative) and G1-phase (geminin- and PCNA-negative) were determined.

The data show that both, *T. gondii wt* and HCE1-deficient strain infection arrest the host cells in S-phase whilst the MYR1-deficient strain fails to do so. The scale bar represents 5 μm. Graph bars represent the median ± SD of three biological donors. The *p* values were calculated using a one-way ANOVA followed by a Kruskal Wallis multiple comparison test of variance, using the non-infected condition (n. i.) as control. *p* < 0.05 was considered as significant.

**Fig. 2 | *T. gondii* infection induces several hallmarks of genome instability in primary human endothelial cells.** HUVEC (*n* = 3) were infected with tachyzoites (MOI 0.5:1) of *TgΔhce*1, *TgΔmyr*1 mutants and control parasites (*TgΔku*80) (**A**) Comet assay was used to detect parasite-driven general DNA strand breaks induction (at 12 h p. i.) and showed that *T. gondii* infection induced DNA damage in a MYR1- and HCE1-independent fashion. Scale bar: 100 μm. **B, C** Host cell genome instability was evaluated by cell binucleation (indicating cytokinesis failure) and micronuclei formation. Current data show an increased percentage of binucleated cells and micronuclei in HUVEC infected with all, *TgΔhce*1, *TgΔmyr*1 mutants and control parasites (*TgΔku*80). **D** Exemplary illustrations of a *T. gondii*-infected host cell experiencing binucleated phenotype, and cells with micronuclei and simultaneous staining of γH2AX-based DNA damage foci. The results show that most of the binucleated cells were positive for DNA damage in contrast to micronuclei. Micronuclei are indicated by white arrowheads. On average, 1400 cells were counted in each non-infected and *T. gondii* strain infection in (**A**−**C**). The scale bar represents 5 μm. Graph bars represent the median ± SD of three biological donors. The *p* values were calculated using a one-way ANOVA followed by a Kruskal Wallis multiple comparison test of variance, using the non-infected condition (n. i.) as control. A *p* < 0.05 was considered as significant.

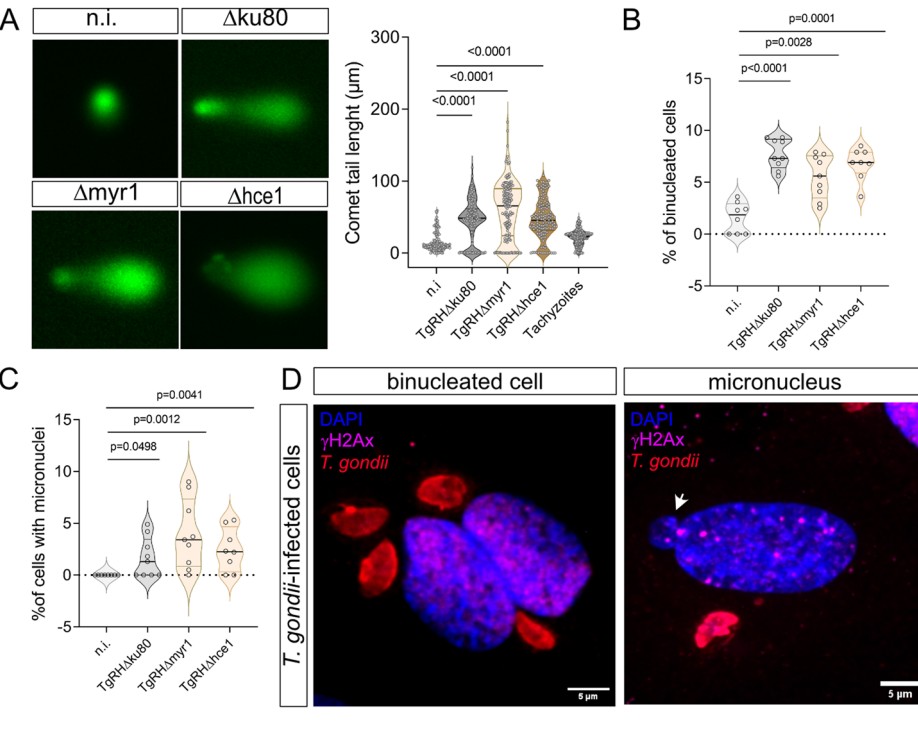

Recently, the upregulation of cyclin E1 by *T. gondii* was suggested to be linked to an S-phase block exit in *T. gondii*-infected host cells[12,13]. In order to analyse whether *T. gondii*-triggered host cell cycle S-phase arrest in HUVEC was related to host cell cyclin E1 overexpression and how early in the parasite intracellular development occurred, we focused on analysing the role of MYR1 and HCE1 in inducing S-phase arrest, by performing infections with *TgΔmyr*1- and *TgΔhce*1 tachyzoites. Overall, HCE1 depletion in tachyzoites proved irrelevant for S-phase arrest, since almost the same kinetic of S-phase stasis was detected in *TgΔhce*1-infected HUVEC and *TgΔku*80-infected control cells (Fig. 1B). In contrast, infections with *TgΔmyr*1 tachyzoites failed to induce S-phase host cell cycle arrest (Fig. 1B) thereby indicating that MYR1 itself or indirect effects of MYR1 are essential for cell cycle impairment. Taken together, the current data suggested that *T. gondii* induces host cell cycle arrest in the S-phase as soon as 15 min p. i. by a mechanism which seems partially controlled by MYR1.

### *T. gondii* infection induces hallmarks of genome instability in primary human umbilical vein endothelial cells and drives the formation of DNA damage foci in a MYR1-dependent manner

During S-phase, cells duplicate their genomic DNA ensuring that daughter cells will receive the same genetic information as the mother cell. Therefore, progression through this cell cycle phase is highly controlled by several checkpoints detecting any DNA damage. To test if *T. gondii*-driven arrest in S-phase is linked to host cell DNA damage, HUVEC were analysed for DNA damage at 12 h p. i. by conventional "comet assays" to study the general integrity of host cellular DNA strands. In an electric field, fragmented DNA runs faster than intact DNA strands, thereby forming a 'comet tail'-like phenotype and becoming longer with the number of strand breaks. Three HUVEC donors were infected with all *T. gondii* strains (*TgΔmyr*1, *TgΔhce*1 and *TgΔku*80) for 24 h. At this time, the samples were incubated in an alkaline solution to reveal DNA-strand breaks and stained with a green, fluorescent probe that binds DNA. After the samples were submitted to an electric field, several random pictures were taken, and the length of the green signal was measured (Fig. 2A). Here, comet length quantification showed that *T. gondii*-infected host endothelial cells suffer from DNA damage driven by a significant increase in DNA strand breaks in comparison to non-infected controls (Fig. 2A). This effect occurred independent of MYR1 and HCE1 since all strains (*TgΔmyr*1, *TgΔhce*1 and *TgΔku*80) induced DNA damage and did not differ significantly in their effects (Fig. 2A).

Recently, *T. gondii* infections of primary endothelial host cells were demonstrated to induce centrosome overduplication, chromosome misallocation outside of the metaphase plate and chromosome missegregation[10,14]. When combined with DNA damage; these findings strongly indicate that *T. gondii* induces genome instability in primary host endothelial cells. In general, genome instability refers to the failure to maintain genome integrity and susceptibility to genetic alterations, including gene mutations, DNA copy number variations, chromosomal rearrangements and aneuploidy[21]. To assess other hallmarks of genome instability than DNA damage, we also quantified host cell binucleation (= cytokinesis failure) and the formation of micronuclei in HUVEC early after infection (3 h p. i.) using *TgΔmyr*1, *TgΔhce*1 and control (i.e., *TgΔku*80) strains. Therefore, HUVEC were stained with DAPI as nuclear/DNA marker and β-catenin as plasma membrane marker to assess if, in case of binucleation, both nuclei indeed belong to the identical cell. Overall, current data revealed a significant increase in host cell binucleation in *TgΔku*80-infected host endothelial cells when compared with uninfected controls (Fig. 2B, D). These effects proved MYR1- and HCE1-independent since similar results were observed for *TgΔmyr*1- and *TgΔhce*1 infections (Fig. 2B). Also, we analyzed the formation of micronuclei in *T. gondii*-infected HUVEC at 3 h p. i. Micronuclei are DNA-positive small membrane-bounded compartments located close to but separated from the cell nucleus (exemplary illustration in Fig. 2D, white arrowheads) which result in a loss of genetic material during mitosis. Even though on a low level, infections with *T. gondii* tachyzoites indeed induced an increase in micronuclei formation (Fig. 2C, D), thereby supporting the hypothesis of parasite-driven genome instability.

**Fig. 3 | *T. gondii* infection induces MYR1-dependent DNA damage foci early after infection.**
**A**, **B** HUVEC (*n* = 3) were infected with tachyzoites (MOI 0.5:1) of *Tg*Δ*hce*1, *Tg*Δ*myr*1 mutants and control parasites (*Tg*Δ*ku*80), fixed at 15 min, 1 and 3 h p. i. and stained for γH2AX (cyan) to quantify DNA damage foci. Tachyzoites were detected by specific antibodies (red) and the nuclei were stained by DAPI (blue). **B** The percentage of DNA damage-positive cells was quantified in comparison to the total number of cells in the field of view at 15 min, 1 h and 3 h p. i. Data show that *T. gondii* infection-driven double-strand DNA breaks occur already at 15 min p. i. in a MYR1-dependent manner. The p-values were calculated using a one-way ANOVA followed by a Kruskal Wallis multiple comparison test of variance, using the non-infected condition (n. i.) as control. A *p* < 0.05 was considered as significant.

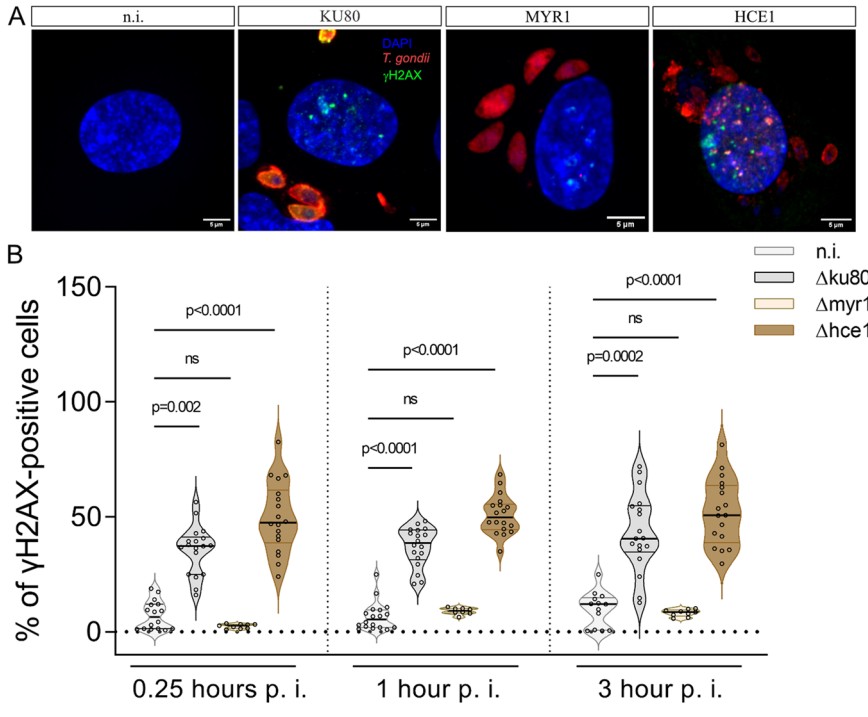

However, this finding also proved independent from MYR1 or HCE1 since *Tg*Δmyr1- and *Tg*Δhce1 infections also resulted in an increased proportion of HUVEC with micronuclei (Fig. 2C).

To estimate the onset of parasite-driven DNA damage during early infection, *T. gondii*-infected human host endothelial cells were stained for γH2AX (histone H2A, phosphorylated at Ser139), which is a sensitive and early marker of DNA double-strand breaks[22]. In case of DNA damage, γH2AX accumulates in the nucleus thereby forming reversible DNA damage foci correlating with the extent of DNA double-strand breaks induction. *T. gondii*-infected host endothelial cells were analyzed for γH2AX-based DNA damage foci formation at 15 min (=0.25 h), 1 and 3 h p. i. (for exemplary illustration, please refer to Fig. 3A). Interestingly, we found a significant increase in DNA damage foci in *T. gondii*-infected HUVEC as early as 15 min p. i. and showed that this effect was maintained until the end of experimentation (3 h p. i.) (Fig. 3B). Overall, parasite-mediated DNA double-strand breaks induction proved MYR1- but not HCE1-dependent, since - in contrast to *Tg*Δhce1 infections - the number of γH2AX-positive host cells did not increase in *Tg*Δmyr1-infected HUVEC (Fig. 3B). DNA damage was equally detected in non-infected and infected cells in the infected monolayer (Supplementary Fig. S1B). These data suggest that *T. gondii*-driven host cellular DNA damage is either related to MYR1-based parasite protein export or the modulation of *c-myc* activity rather than to the modulation of cyclin E1 expression. To furthermore link the presence of DNA damage foci to the actual cell cycle phase, we additionally performed a co-staining of γH2AX (DNA damage foci) and cell cycle phase markers (S-phase marker: PCNA, G2-phase marker: geminin) in *Tg*Δmyr1-, *Tg*Δhce1- and *Tg*Δku80-infected HUVEC at 3 h p. i. (for exemplary illustration see Fig. 4A). Overall, cells experiencing DNA damage foci were in principle found in all cell cycle phases, i.e., in G1, G2 and S-phase, with the overall highest number of DNA-damaged host cells to be found in the G2-phase. As an interesting finding, a MYR1-dependent absence of DNA damage foci only applied to host cells in S-phase thereby restricting MYR1-driven effects to the major phase of DNA synthesis (Fig. 4B). Taken together, the current data indicate that *T. gondii* tachyzoite infection induces DNA damage in host cells as early as 15 min p. i. depending on MYR1 activity.

## *T. gondii* infection triggers the activation of the ATM-dependent pathway of DNA damage response

In general, eukaryotic cells maintain genome integrity insults driven by endogenous or environmental genotoxic stress by activating DNA damage response (DDR) pathways, thereby allowing cells to rapidly repair different types of DNA damage. Therefore, DDR pathways also control cell cycle progression to guarantee correct genomic DNA synthesis before allowing cells to proceed into mitosis[23,24]. In case of DNA damage, cells may activate the homologous recombination repair (HR) pathway, which is restricted to the S-phase of the cell cycle. Depending on the type of DNA damage, i.e., single- and double-strand breaks, two different branches of this pathway, i.e., the ataxia telangiectasia and Rad3 related (ATR)- or the ataxia-telangiectasia mutated (ATM)-pathway, respectively, are then activated. Given that *T. gondii*-infected host cells were arrested in the S-phase, we hypothesized that the HR repair pathway was indeed activated and here focused on an alteration of molecules related to ATR- and ATM pathways in *Tg*Δmyr1-, *Tg*Δhce1- and control (*Tg*Δku80)-infected HUVEC. Therefore, the expression of different key proteins was analysed in *T. gondii*-infected HUVEC at 12 h p. i. by Western blotting (WB). Overall, a significant upregulation of the ATM pathway-related sensor proteins Rad50 and ATM was found in *Tg*Δku80-infected HUVEC indicating that *T. gondii* control infections indeed triggered an HR-based DDR in HUVEC (Fig. 5 and Supplementary Fig. S2). These reactions proved independent of MYR1 and HCE1 since infections with *Tg*Δmyr1 and *Tg*Δhce1 strains upregulated Rad50 and ATM expression equally to controls (Fig. 5). However, when examining effectors and other downstream proteins of the repair pathway, exclusively CDK2 was found downregulated by WB in the case of *Tg*Δmyr1 and *Tg*Δhce1 infections whilst p21 and p53 expression was not affected by parasite infection (Fig. 5). In line with previous reports[7], infections with the control strain (*Tg*RHΔ*ku80*) in principle induced host cyclin E1 upregulation at 12 h p. i., however, these reactions did not reach a statistical significance level in the current study (*p* = 0.15, Fig. 5). Taken together, the current data indicated that primary HUVEC reacted to *T. gondii*-tachyzoite infections by a DDR based on HR pathway activation, specifically via ATM kinase induction.

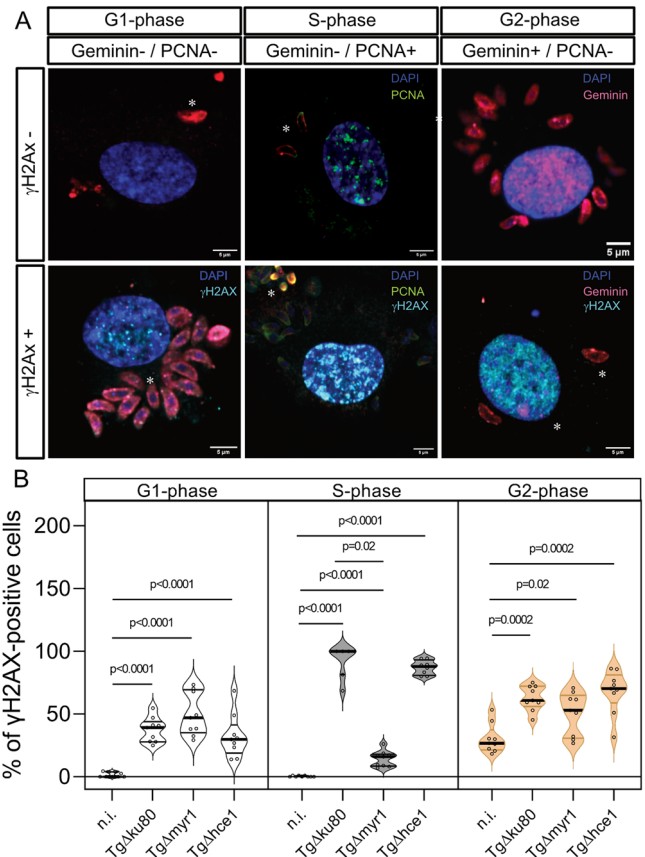

**Fig. 4 | *T. gondii*-driven DNA damage in relation to host cell cycle phases.**
**A, B** HUVEC (*n* = 3) were infected with tachyzoites (MOI 0.5:1) of *TgΔhce*1,
*TgΔmyr*1 mutants and control parasites (*TgΔku*80). **B** *T. gondii*-infected cells
(3 h p. i.) were stained in parallel for γH2AX (DNA damage foci marker), geminin
and PCNA to study whether DNA damage foci formation was linked to a distinct cell
cycle phase. γH2AX-positive cells were assigned to each cell cycle phase and plotted
as the percentage of DNA damage-positive cells. The scale bar represents 5 μm.
Graph bars represent the median ± SD of three biological donors. The p-values were
calculated using a one-way ANOVA followed by a Kruskal Wallis multiple com-
parison test of variance, using the non-infected condition (n. i.) as control. A *p* < 0.05
was considered as significant.

### *T. gondii*-induced host cellular DNA damage is ROS-independent

Given that increased ROS production has been described as one of the major
cellular drivers of DNA damage, we here estimated if *T. gondii*-tachyzoite
infection triggers intra- or extracellular ROS production in HUVEC within
3 h of infection. Intracellular ROS was measured in DCFH-DA pre-incu-
bated cells whilst extracellular ROS was detected by the Amplex Red reagent.
After 3 h p.i., the cells were loaded with DCFH-DA or Amplex red and
analyzed by FACS-based quantification or luminometry, respectively. Non-
infected cells were used as the control for infection whilst non-stained cells
were used as the assay control (WOS: without staining). Overall, neither
intra- nor extracellular ROS production was induced by parasite infection
(Fig. 6A−C). Notably, intracellular ROS concentrations were even sig-
nificantly lower in *T. gondii*-infected HUVEC compared to non-infected
controls at 3 h p. i. (Fig. 6A, B). To demonstrate that *T. gondii* indeed reduces
intracellular ROS production, we pre-incubated cells with the ROS inhi-
bitor, NAC (N-acetyl-l-cysteine). As we expected, this compound was able
to block intracellular ROS generation in non-infected cells and also in those
cells that after treatment were infected with *T. gondii*, thereby confirming
that *T. gondii*-tachyzoite infection indeed does not induce a ROS-based
reaction in HUVEC. To demonstrate that ROS was not involved in parasite-
driven DNA double-strand breaks induction, we pre-treated host cells with
NAC previously to perform the infection. After 3 h, the cells were fixed and

stained for γH2AX to analyse DNA damage foci formation. The results
showed that the inhibition of ROS production before infection did not
prevent cells from displaying DNA damage induction by *T. gondii* given that
the percentage of DNA damage foci was similar to cells without NAC
treatment (Fig. 6D). In agreement with ROS-related findings mentioned
above, it was confirmed that host cellular DNA damage occurs in a ROS-
independent manner and is driven by *T. gondii*-tachyzoite infection, thereby
excluding ROS-based oxidative responses as a mechanism of *T. gondii*
infection-mediated DNA damage at 3 h p. i.

## Discussion

Besides other essential cell functions, the zoonotic-relevant parasite *T. gondii*
is well recognised to alter the host cell cycle. Recent studies demonstrated
that several hallmarks of genome stability are seriously affected by infection,
resulting in chromosome displacement, the formation of multipolar spin-
dles, aberrant mitosis and cytokinesis failure irrespective of *T. gondii*-hap-
lotypes (I, II, III) used[14,18]. Consistently, a recent report revealed infection-
driven DNA damage based on DNA double-strand breaks in *T. gondii*-
infected tumoral cell lines (HeLa) and identified ROS generation as a major
driver of this DNA damage[25]. In the current study, we aimed to study the
early onset of *T. gondii*-driven DNA damage in primary human host
endothelial cells to be as close as possible to the in vivo situation[18], and more
importantly intended to analyse the role of the parasite-derived effector
molecules HCE1 and MYR1 in *T. gondii*-triggered DNA damage. To create
a rather physiological scenario, we here avoided cell cycle alterations due to
an immortalised host cell status and potentially overlapping effects driven by
simultaneous parasite replication or by parasite overload (when applying
high MOIs) as previously reported[14,18]. Thus, we studied the early phase of *T.
gondii*-tachyzoite infection (15−180 min p. i., i.e., the timeframe before
parasite mitosis occurs), used primary human endothelial cells of low pas-
sages as host cells and applied a low MOI for host cell infection.

 *T. gondii* tachyzoites are well-known in literature to stall the host cell
cycle between the S-phase and G2/M transition. Nonetheless, almost all
previous studies investigated rather later time points of infection, in which
the first round of intracellular parasite duplication has already been
accomplished[11–13,26]. Thus, *T. gondii*-infected human foreskin fibroblasts
(HFF) were arrested in the G1- to S-phase transition at 24 h p. i. whilst
human trophoblasts or dermal fibroblasts showed an arrest in the G2/M
phase at 8 or 24 h p. i.[11,13,27]. Bovine-, rat- and human-derived host cells were
shown to be arrested at 24 and 48 h after *T. gondii* tachyzoite infection in the
G2/M- or S-phase, respectively[14,26]. Considering these findings, host cell
cycle control seems a general strategy of this polyxenous apicomplexan
parasite infecting any kind of warm-blooded host species or nucleated host
cell types. However, the precise cell cycle phase, in which the host cell is
arrested seems to differ between cell types. By focusing on early wt *T. gondii*
infection of primary human endothelial cells (HUVEC), we here demon-
strate that the onset of cell cycle arrest occurs strikingly early after parasite
invasion and is already stably established at 15 min p. i. Thus, HUVEC
showed a significant stasis in S-phase at 15 min p. i. and this effect was
maintained until the end of experimentation (3 h p. i.). On a mechanistic
level, a recent report revealed *T. gondii*-driven S-phase arrest to be linked to
an incapability of infected host cells to integrate new DNA molecules
thereby implying a hindrance in DNA synthesis[12]. Concomitant with early
S-phase arrest, we also detected a significant induction of DNA double-
strand breaks in wt *T. gondii*-infected cell layers indicating that this insult is
most probably linked to S-phase arrest. DNA double-strand damage is one
of the most deleterious events in biology and may lead to further long-term
consequences such as genetic disorders or even tumours. In some host cells,
this may manifest in host cell endocycling and DNA duplication resulting in
cytokinesis failure. In line and agreement with earlier reports[10,14,15], we here
demonstrate a significantly enhanced level of binucleation in *T. gondii*-
infected human host endothelial cells. Interestingly, we additionally
observed that binucleated host cells tended to be DNA damage foci-positive
in both cellular nuclei suggesting that cytokinesis failure could be related to
DNA strand damage during S-phase. It is known that DNA damage is

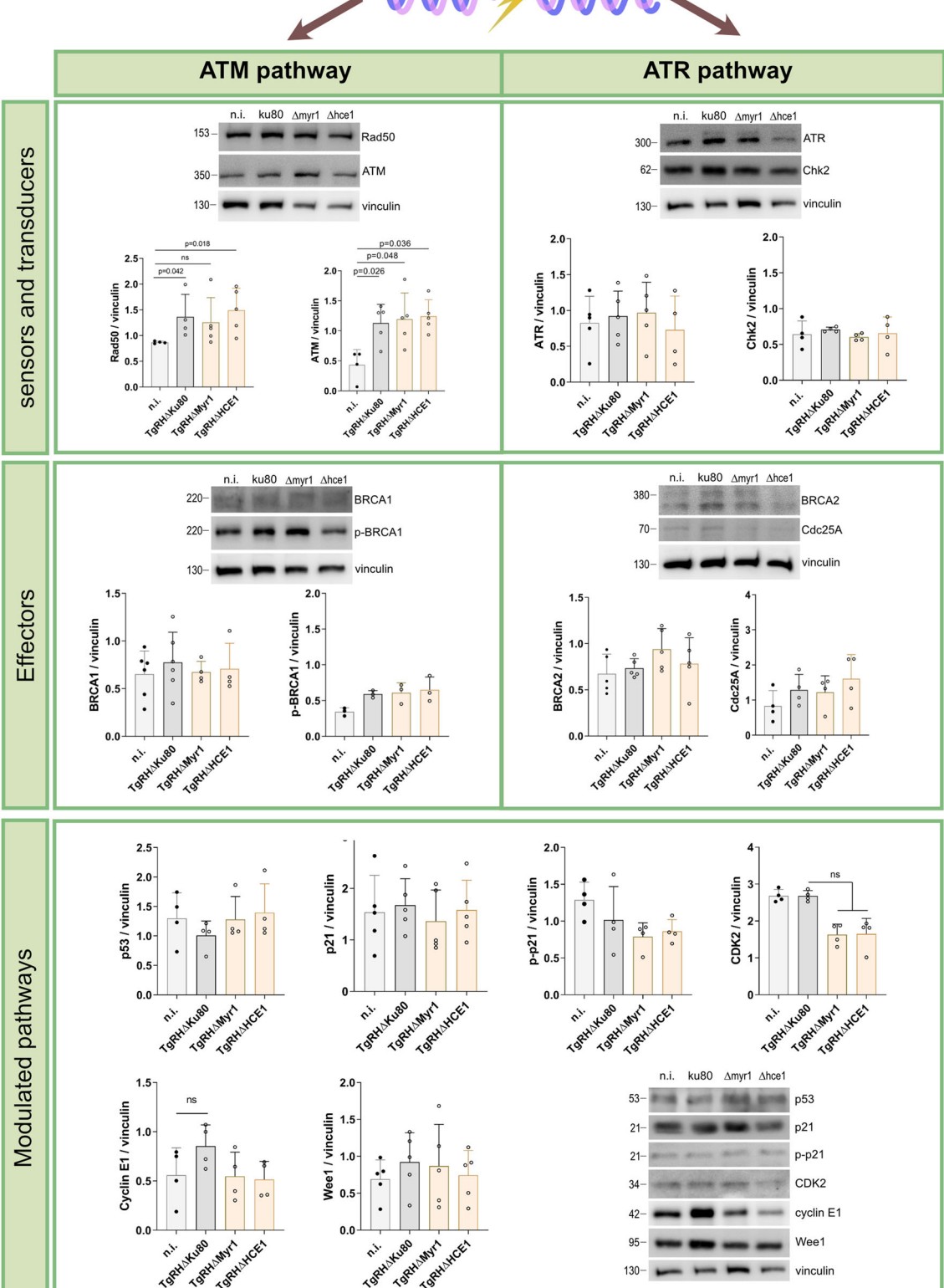

**Fig. 5 | *T. gondii* infection activates the ATM-dependent DNA repair response pathway in primary human host cells.** HUVEC were infected with tachyzoites (MOI 0.5:1) of *TgΔhce*1, *TgΔmyr*1 mutants and control parasites (*TgΔku8*0) and analysed for key molecules of the ATM or ATR pathway by Western blotting-based protein quantification at 12 h p. i. Graph bars represent the median ± SD of four to six biological replicates. Vinculin quantification was used as a loading control of the assay. The results show that exclusively the ATM-dependent pathway was activated by *T. gondii*-infection in a MYR1- and HCE1-independent fashion. The *p* values were calculated using a one-way ANOVA followed by a Kruskal Wallis multiple comparison test of variance, using the non-infected condition (n. i.) as control. A *p* < 0.05 was considered as significant.

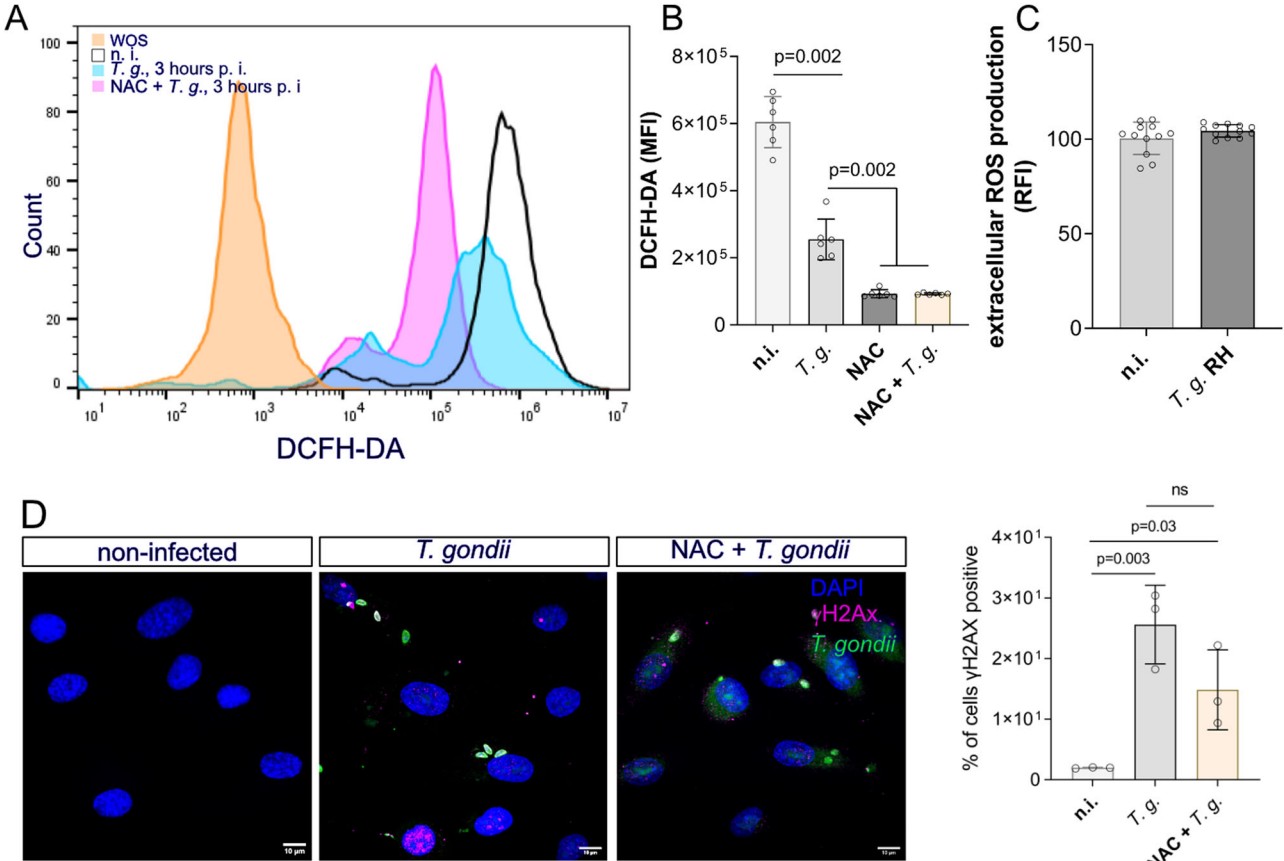

**Fig. 6 | *T. gondii*-driven DNA damage in HUVEC is ROS-independent.** HUVEC (*n* = 6) were infected with control tachyzoites (MOI 0.5:1) and both intracellular and extracellular ROS production was evaluated after 3 h p. i. **A** Exemplary histogram using flow cytometry-based quantification of intracellular ROS via the probe DCFH-DA. NAC treatment was used to block ROS production. DCFH-DA-based quantification of intracellular ROS production in *T. gondii*-infected HUVEC in the presence and absence of the ROS inhibitor NAC. A control of cells without DCFH-DA staining was included (WOS: without staining). Data showed that *T. gondii*-infected cells produced lower levels of ROS than uninfected controls. MFI: mean of fluorescence. FACS gating strategy in Supplementary Fig. S4. **B** Quantification of extracellular ROS (Amplex red) in *T. gondii*-infected HUVEC and control cells showed no infection-driven changes. RFI: relative fluorescence intensity. **C** Quantification of γH2AX-based DNA damage foci in NAC-treated and untreated *T. gondii*-infected HUVEC. No changes in DNA damage induction were detected when cells were pre-treated with the ROS inhibitor (NAC), suggesting that *T. gondii*-driven DNA damage occurs ROS-independently. The *p* values were calculated using a one-way ANOVA followed by a Kruskal Wallis multiple comparison test of variance, using the non-infected condition (n. i.) as control. A *p* < 0.05 was considered as significant.

commonly observed in cells undergoing aberrant mitosis due to failures during cytokinesis leading to a bi/multinucleated phenotype[28]. Besides cytokinesis failure, we additionally showed *T. gondii* wt infection-driven formation of both micronuclei and aberrant mitotic spindles. The sum of all these alterations may severely affect host cell ploidy (Scheme 1), resulting in a cellular status of 4n-8n, as recently demonstrated for *T. gondii* RH strain-infected RAW264.7 cells[15]. Undoubtedly, all these findings represent hallmarks of genome instability, which may develop as a consequence of the initial, *T. gondii*-driven DNA insult.

Since DNA damage represents a severe event for cell integrity, different pathways of DNA damage repair have evolved to prevent potential deleterious long-term consequences. Given that *T. gondii* wt infections induce S-phase arrest and DNA double-strand breaks, we here studied the activation of the HR repair pathway in HUVEC early after *T. gondii* infection. In line with former reports on other host cell types[25,29,30], *T. gondii*-infected HUVEC showed activation of the ATM-branch of the HR pathway, which classically responds to DNA double-strand insults, thereby indirectly supporting current findings on enhanced DNA damage foci formation. However, when considering that S-phase arrest and DNA damage foci are maintained in *T. gondii*-infected host cells over time, it seems likely that host cell attempts to repair DNA damage are failing and leading to different hallmarks of genome instability as described above.

So far, the exact trigger of *T. gondii*-driven host cell DNA damage in primary HUVEC or BUVEC remains unknown but seems clearly worthy of future investigations as previously suggested[14,18]. However, *T. gondii* infections are well-documented to affect host cellular mitochondrial function and integrity[8]. In general, aerobic cells mainly generate energy via oxidative respiration in mitochondria with ROS being synthesised as a byproduct. ROS have long been implicated as highly toxic molecules for cells, consequently, intracellular ROS concentration is constantly controlled to avoid macromolecule damage. Also, DNA is vulnerable to ROS and high ROS concentrations are well-documented as triggers of double-strand breaks induction[29]. Therefore, we here monitored intra- and extracellular ROS production by *T. gondii*-infected HUVEC. In contrast to our expectations, intracellular ROS production was even decreased, and extracellular ROS levels remained unchanged by infection thereby denying any ROS-related effects on DNA integrity. In line, inhibition of ROS generation by NAC treatments did not change the formation of DNA damage foci in *T. gondii*-infected HUVEC. The ROS-related findings are in principle in agreement with data on *T. gondii*-infected ARPE-19 cells[31] but contrast with a previous report on *T. gondii*-infected immortalised cell lines, which identified ROS generation as the major driver of DNA damage (detected at 24 h p. i.)[25]. The discrepancy with the latter results may rely on several aspects: the different time points post-infection, the different host cell types (tumoral vs primary) and the 20-fold higher ratio of infection (1:10 vs 1:0.5 of this study). Given

**Scheme 1 |** Hypothetic illustration of *T. gondii*-driven effects on host cellular genome integrity.

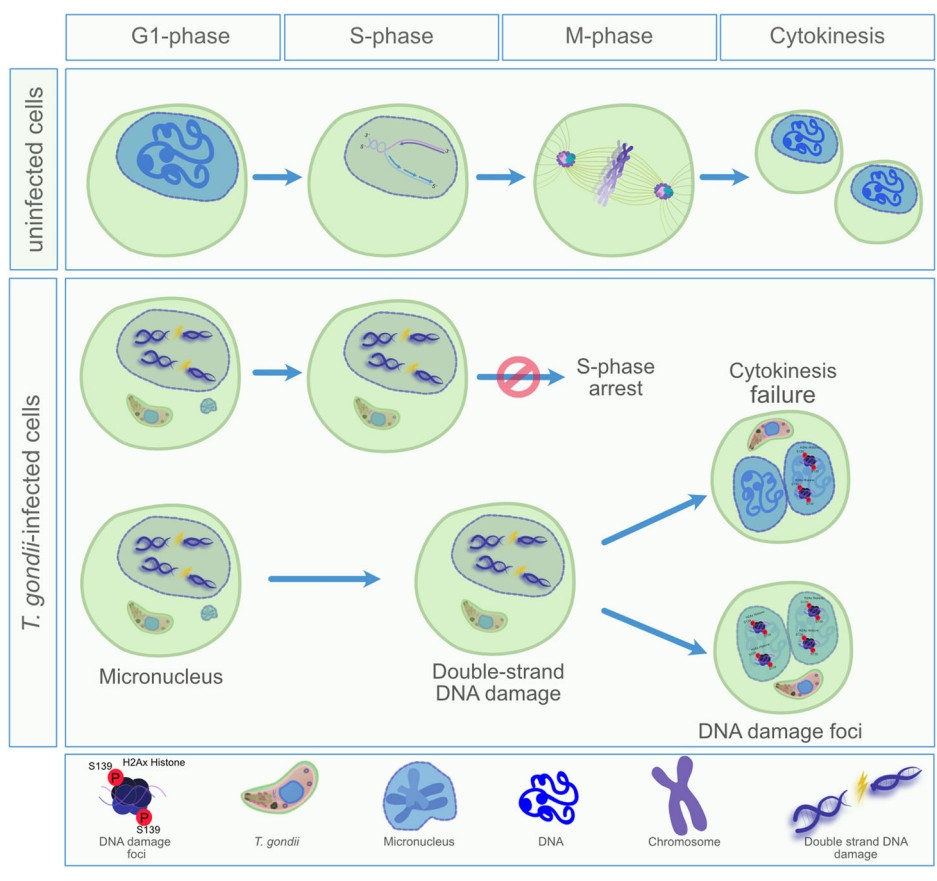

that available data on *T. gondii*-driven ROS synthesis in host cells are inconsistent, ROS-related effects may simply rely on the cell type used for infection.

Considering the very early effects of *T. gondii* infection on cell cycle stasis and DNA damage foci induction, it seems very likely that these effects are induced by parasite-derived molecules. It is well-documented that *T. gondii* tachyzoites secrete a variety of effectors with several of them being translocated through the PVM and acting on signalling cascades of the host cell. The translocation process from the PV to the host cell cytosol or nucleus seems to mainly rely on a multiprotein complex being located in the PVM. MYR1 represents one key element of this complex and was proven to control the key transcription factor *c-myc* in host cells (ref. [5], p. 16). Interestingly, Naor et al.[32] reported that a substantial proportion of transcriptional host gene changes during *T. gondii* infection is MYR1-dependent. Amongst these, gene sets associated with E2F and c-Myc transcription factors, G2/M checkpoint, and DNA repair demonstrated a dependence on MYR1. MYR1 was also proven as pivotal for GRA16 translocation from the PV membrane into the host cell nucleus[5], where it modulates genes involved in cell cycle progression and the p53 tumour suppressor pathway. Moreover, the parasite-derived effector molecule HCE1, which was shown to upregulate the key cell cycle regulator cyclin E1, proved as MYR1-dependent[7,15]. All these findings rendered MYR1 and HCE1 as good candidates for parasite-driven effects on cell cycle stasis and DNA damage. Therefore, we performed related analyses using *Tg*Δmyr1 and *Tg*Δhce1 mutants. In line with previous findings on a failure of MYR1-deficient tachyzoites to arrest the host cell cycle at 19 h p. i. in HFF cells[15], we here showed that in - contrast to control infections - *Tg*Δmyr1 tachyzoites were unable to arrest HUVEC in S-phase early after infection (15−180 min p. i.). This finding may either mirror MYR1-driven effects on *c-myc* expression or indirectly be mediated by one or several of the various MYR1-dependent parasite effectors. However, this finding seems not to be dependent on cyclin E1 expression since infections with a *Tg*Δhce1 mutant-

induced S-phase arrest are equal to control infections. We furthermore showed that *Tg*Δmyr1- but not *Tg*Δhce1-tachyzoites failed to induce DNA double-strand breaks in host cells, thereby supporting the hypothesis of DNA damage as an initial trigger of parasite-driven S-phase stasis. However, when detecting general DNA damage in infected cell layers by comet assays, infections with *Tg*Δmyr1, *Tg*Δhce1 and *Tg*Δku80 strains all significantly induced DNA damage. Currently, we do not have an explanation for this conflicting result, however, it may be because comet assays detect any kind of DNA strand damage and cannot discriminate single- and double-strand breaks-based insults. Thus, other MYR1-independent mechanisms of cell cycle progression impairment may also be triggered by *T. gondii* in general. In line, host cell binucleation, micronuclei formation and chromosome missegregation were also detected in both *Tg*Δmyr1- and *Tg*Δhce1-infected HUVEC proving these effects as MYR1- and HCE-independent.

In summary, current data indicated that *T. gondii* tachyzoite infection controls host cell cycle progression in HUVEC very soon after infection (15 min p. i.) thereby inducing S-phase stasis with concurrent, ROS-independent DNA double-strand breaks induction both of which depend on functional parasite-derived MYR1 but not on HCE1 expression. These cellular insults bear long-term consequences, such as the formation of micronuclei, aberrant mitosis and binucleation thereby indicating that host cell attempts to clear DNA insults via an ATM-related DDR may finally fail in some cells (Table 1). However, understanding the underlying mechanism needs further studies.

In conclusion, while the results presented in the current work indicated that DNA damage occurs early and in a MYR1-dependent manner, resulting in S-phase arrest, the underlying reasons why *T. gondii* damage the host cell DNA remains unexplored. This raises critical questions for future research: what is the parasite's advantage in inducing DNA damage? Does DNA damage lead to S-phase arrest, facilitating the parasite's survival, or is DNA damage a result of S-phase arrest driven by effector protein export? Addressing these questions will be essential in further understanding the

**Table 1 | Antibodies used in the current study**

| Antigen | Company | Cat. number | Origin/reactivity | Dilution |
|---|---|---|---|---|
| **Primary antibodies IFA** | | | | |
| Histone H3 (S10) | Abcam | Ab5176 | Rabbit | 1:300 |
| H2AvD (pS139) | Cell Signalling | 9718S | Rabbit | 1:200 |
| Histone H2A.X (pS139) | Cell Signalling | 80312S | Mouse | 1:400 |
| PCNA | Abcam | ab18197 | Rabbit | 1:200 |
| Geminin | Cell Signalling | 52508 | Rabbit | 1:800 |
| *Toxoplasma gondii* | Thermo Fisher | PA1-7256 | Goat | 1:100 |
| β-catenin | Abcam | ab32572 | Rabbit | 1:200 |
| **Primary antibodies WB** | | | | |
| Vinculin | Santa Cruz | sc-73614 | Mouse | 1:1000 |
| HR DNA repair antibody sampler kit | Cell Signalling | 99891 | Rabbit | 1:1000 |
| MNR complex Antibody Sampler kit | Cell Signalling | 8344 | Rabbit | 1:1000 |
| **Secondary antibodies** | | | | |
| **Antigen/conjugate** | **Company** | **Cat. number** | **Host/target** | **Dilution** |
| **Goat anti-mouse IgG Peroxidase conjugated** | | | | |
| Goat anti-rabbit IgG Peroxidase conjugated | Pierce | 31430 | goat/mouse | 1:40,000 |
| Goat anti-mouse IgG Peroxidase conjugated | Pierce | 31460 | goat/rabbit | 1:40,000 |
| Alexa Fluor 488 | Thermo Fisher | A11001 | goat/mouse | 1:500 |
| Alexa Fluor 488 | Thermo Fisher | A11008 | goat/rabbit | 1:500 |
| Alexa Fluor 594 | Thermo Fisher | A21468 | chicken/goat | 1:500 |
| Alexa Fluor 647 | Thermo Fisher | A21244 | Goat/rabbit | 1:500 |
| Alexa Fluor 647 | Thermo Fisher | A21235 | Goat/mouse | 1:500 |

**Table 2 | Primary cells and isolates used in the current study**

| Name of primary cell | Company | Cat. number | Lot number (donor) |
|---|---|---|---|
| HUVEC-p single donor | Promo Cell | C-12250 | 449Z018.1 |
| HUVEC-p single donor | Promo Cell | C-12250 | 466Z026 |
| HUVEC-p single donor | Promo Cell | C-12250 | 467Z015 |
| HUVEC-p single donor | Promo Cell | C-12250 | 478Z023 |
| HUVEC-p single donor | Promo Cell | C-12250 | 486Z004 |
| HUVEC-p single donor | Promo Cell | C-12200 | 469Z003.1 |
| HUVEC-p single donor | Promo Cell | C-12200 | 473Z022.1 |
| HFF-1 | ATCC | SCRC-1041 | |

intricate interactions between *T. gondii* and its host, potentially revealing new targets for therapeutic intervention.

## Materials and methods
### Primary human host cells and parasite maintenance
Primary human umbilical vein endothelial cells (HUVEC, $n = 6$, Promocell), were cultured at 37 °C and 5% $CO_2$ following the supplier's protocols (see Table 2). Each experiment was performed at a maximum of four passages after isolation to avoid aging-driven artefacts. All HUVEC isolates were seeded at the same time and infected by the same batch of tachyzoites. *T. gondii* tachyzoites (RH, Δ*ku*80, Δ*myr*1 and Δ*hce*1 strain) were maintained by serial passages in primary HFF cells (ATCC, maximum passage: 10). Therefore, free-released *T. gondii* tachyzoites were harvested from HFF supernatants, pelleted ($400 \times g$, 12 min), counted in a Neubauer chamber and suspended in the corresponding medium for each host cell type. Infections were performed in subconfluent cell layers [immunofluorescence assays and ROS measurements: 6−12-well formats; immunoblotting and FACS assays: T-25 cm² flask format (Greiner); Western blot quantification:

T-75 cm² flasks (Greiner)]. All experiments were performed at an MOI of 1:1 or 0.5:1 (parasites: cells).

### Protein extraction, SDS-PAGE, and immunoblotting
Proteins were extracted from *T. gondii*-infected and non-infected HUVEC by cell sonication (20 s, 5 times) in RIPA buffer (50 mM Tris-HCl, pH 7.4; 1% NP-40; 0.5% Na-deoxycholate; 0.1% SDS; 150 mM NaCl; 2 mM EDTA; 50 mM NaF, all Roth) supplemented with a protease inhibitor cocktail (1:200, Sigma-Aldrich). Cell homogenates were centrifuged ($10.000 \times g$, 10 min, 4 °C) to sediment intact cells, nuclei and detritus. Respective supernatants were collected and estimated for protein content via the BCA Assay Kit (Thermo Scientific) following the manufacturer's instructions.

For immunoblotting, protein samples were supplemented with Laemmly-β-mercaptoethanol loading buffer (4x Laemmli Sample Buffer, 1610747, BioRad). After boiling (95 °C) for 5 min, proteins (40 µg/slot) were separated in 10−15% polyacrylamide gels via electrophoresis (100 V, 1.5 h; tetra system, BioRad) and then transferred to polyvinylidene difluoride membranes (Millipore) (300 mA, 2 h at 4 °C). Samples were blocked in 3% BSA in TBS [50 mM Tris-Cl, pH 7.6; 150 mM NaCl containing 0.1% Tween (blocking solution); Sigma-Aldrich] for 1 h at room temperature (RT) and then incubated with primary antibodies (Table 1) diluted in blocking solution (overnight, 4 °C). Detection of vinculin was used as loading control for sample normalisation. Following three washes in TBS-Tween 0.1% buffer, blots were incubated in secondary antibody (Table 1) solutions (dilution in blocking solution, 30 min, RT). Following three further washes in TBS-Tween (0.1%) buffer, signal detection was accomplished by an enhanced chemiluminescence detection system (ECL plus kit, GE Healthcare) and recorded using a ChemoDOC Imager (Bio-Rad). Protein masses were controlled by a protein ladder (PageRuler Plus Prestained Protein Ladder ~10−250 kDa, Thermo Fisher Scientific). Protein band intensities were quantified by the Fiji Gel Analyzer plugin[33].

## Immunofluorescence assays

Cell layers were fixed with paraformaldehyde (4%, 15 min, RT; Roth), washed thrice with PBS and incubated in blocking/permeabilization solution (PBS with 3% BSA, 0.1% Triton X-100; 1 h, RT). Thereafter, samples were incubated in primary antibodies (Table 1) diluted in blocking/permeabilization solution (overnight, 4 °C, in a humidified chamber). After three washes in PBS, samples were incubated in secondary antibody solutions (Table 1; 30 min at RT and in complete darkness). Cell nuclei were labelled by a DAPI-supplemented mounting medium (Fluoromount G, ThermoFisher).

## Immunofluorescence assay for cell cycle phase estimation

Immunofluorescence-based analysis was performed on PFA-fixed cells using antibodies against γH2Ax (DNA damage marker), PCNA (S-phase marker), geminin (G2-phase marker) in addition to DAPI staining for DNA detection. In each experiment, each HUVEC donor was tested in three technical replicates. The number of geminin- or PCNA-positive cells was normalised by the total number of cells in the field of view based on DAPI staining. Given that, isotypes of antibodies used for PCNA and geninin detection were the same, a colocalization of both was not possible, therefore cell cycle phases were quantified in two independent staining assays. The S-phase was calculated in cells stained with PCNA and DAPI. The total number of cells that were positive for PCNA were assigned as cells in the S-phase. For the G1- and G2-phase, cells were stained against geminin and DAPI. Geminin-positive cells were assigned as G2-phase and Geminin-negative in G1-phase. In each case, the percentage was evaluated against the total number of DAPI-positive cells in the same field of view. The parallel assessment of γH2AX-positive cells was used to analyse DNA damage in each cell cycle phase. Thus, DNA-damaged cells in G1 were geminin-/γH2Ax +, in S-phase were PCNA + /γH2Ax +, and in G2-phase were geminin + / γH2Ax +.

## Image acquisition and reconstruction

Fluorescence images were acquired with a ReScan Confocal microscope instrumentation (RCM 1.1 Visible, Confocal.nl) equipped with a fixed 50 µm pinhole size and combined with an Eclipse Ti2-A inverted microscope (Nikon) which included a motorised Z-stage (DI1500, Nikon). The RCM unit was connected to a Toptica CLE laser with the following excitations: 405/488/561/640 nm. Images were taken via a sCMOS camera (PCO edge) using a CFI Plan Apochromat 60x lambda-immersion oil objective (NA 1.4/0.13; Nikon). The setup was operated by the NIS-Elements software (Nikon, version 5.11). Images were acquired via z-stack optical series with a step size of 0.1 microns depth to cover all structures of interest within analysed host cells. Z-series were displayed as maximum z-projections. Identical brightness and contrast conditions were applied for each data set within one experiment using Fiji software[33].

PCNA detection was performed by an automated selection of the nuclear area using Fiji software applying the following workflow: an Otsu threshold was applied to the DAPI channel to obtain the total nuclear area. Particles larger than 800 pixels were selected and merged with the PCNA channel (nuclear selection is exemplary illustrated in Fig. 3 as white circles surrounding the nuclei). The number of host cells in each S-subphase was counted manually according to the instructions given by Schönenberger et al.[34]. For nuclear size analysis, ROIs were measured using Fiji measure plugins following nuclear selection. Cell nuclei were segmented using Otsu thresholding as a binary image. Finally, morphological features (circularity, axes ratio, area, and average intensity) were obtained using particle analysis in Fiji software.

## DNA damage detection by comet assays

Three HUVEC isolates were infected with *T. gondii* tachyzoites at an MOI 1:0.5. At 12 h p. i., non-infected and infected cells were carefully removed from plates by scraping, and transferred to conical tubes. The cell pellet was washed once with ice-cold PBS. Cells were suspended at $1 \times 10^5$ cells/mL in ice-cold PBS and analysed for DNA damage via comet assays (Abcam,

ab238544) according to the manufacturer´s instructions. Briefly, cells ($2.5 \times 10^5$) were resuspended in low melting agarose at 1/10 ratio (v/v) and transferred onto the top of an agarose base layer previously prepared in special glass comet slides, thereby maintaining the cell suspension at 37 °C to avoid gelation. After gelation, cell samples were incubated in 1X lysis buffer [NaCl, EDTA solution, 10X lysis solution (Abcam, ab238544), DMSO] for 1 h at 4 °C to remove cell membranes, cytoplasm and nucleoplasm, and to solubilise nuclear packaging proteins, followed by electrophoresis at 12 V (1 volt/cm according to the chamber used), 240 mA for 30 min at 4 °C under alkaline conditions, washed three times with pre-chilled distilled water for 2 min and once with cold 70% Ethanol for 5 min. In a final step, DNA was visualised by 1X Vista Green DNA staining (15 min at RT), an intercalating DNA dye (Abcam, ab238544). Based on their differential migratory behaviour, intact DNA (= "comet head") can be distinguished from DNA with single-stranded or double-stranded DNA breaks, resulting in "comet tail" structures. Comets (head + tails) were analysed by the OpenComet Software allowing for automated analysis of comet assay images[35]. The DNA damage was quantified by measuring the displacement between intact nuclear DNA (comet head) and the resulting tail, resulting from single- and double-strand DNA breaks. Hence, the tail length (µm) was graphed for this assay.

## Quantification of intra- and extracellular ROS production

Intracellular ROS production was quantified in *T. gondii*-infected HUVEC at 3 h p. i. by incubation in 2,7-dichlorodihydrofluorescein diacetate (DCFH-DA), a cell-permeable probe, which is converted into DCF-DA by esterases present in the cytosol and then oxidised to fluorescent DCF in the presence of intracellular hydrogen peroxide and peroxidases. Here, non-infected and *T. gondii*-infected HUVEC were incubated in 10 µM DCFH-DA for 30 min and thereafter washed with PBS. Cells were trypsinized with Trypsin/EDTA (0.25%), washed twice with PBS and resuspended in PBS. The samples were analyzed by a BD Accuri C6 Plus Flow Cytometer (Becton-Dickinson, Heidelberg, Germany). Cells were gated according to their size and granularity; only morphologically intact host cells were included in the analysis. All analyses were performed by the software FlowJo v.10. Treatments with N-acetyl-L-cysteine, NAC (50 µM NAC, 1 h, 37 °C, ab143032, Abcam) to block ROS production were performed prior to *T. gondii* infection.

Extracellular ROS production was estimated at 3 h p. i. in *T. gondii*-infected HUVEC and non-infected controls using Amplex Red reagent (ThermoFisher). The Amplex Red reagent is a highly sensitive and stable probe for $H_2O_2$ detection, reacting in a 1:1 stochiometry ratio with $H_2O_2$ to generate the fluorescent compound resorufin, which is then detected by spectrofluorescence using 530/590 nm excitation/emission wavelengths, respectively. Cells were incubated for 30 min at 37 °C in a solution containing 41.5 µM Amplex Red and 7.5 U/ml Horseradish Peroxidase (HRP) (ThermoFisher). The resulting fluorescence was measured by a microplate reader (Varioskan Flash, Thermo Fischer).

## Statistics and reproducibility

All data were expressed as median ± SD from at least three independent experiments. The normality of data distribution was assessed by the D'Agostino & Pearson and Shapiro-Wilk tests. When two groups were compared, a Mann−Whitney test was performed. When three or more experimental groups were compared, a Kruskal–Wallis one-way analysis of variance was applied. Significance was defined as $p \leq 0.05$. All graphs and statistical analyses were performed using GraphPad Prism9 software.

## Reporting summary

Further information on research design is available in the Nature Portfolio Reporting Summary linked to this article.

## Data availability

All relevant data are available from the authors upon request (zahady.velasquez@vetmed.uni-giessen.de). WB´s uncropped and unedited blot

images are available in Supplementary Material file (Figure S3). Numerical source data for manuscript graphs are available in Supplementary Data (Raw dat excel file).

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

## Acknowledgements

The authors would like to thank Dr. Moritz Treeck (The Francis Crick Institute, London, UK / Gulbenkian Institute for Molecular Medicine, Porto, Portugal) for providing lab space and access to the TgRHΔku80, TgRHΔmyr1 or TgRHΔhce1 strains. We express our gratitude to Dr. Moritz Treeck, Simon Butterworth (The Francis Crick Institute, London, UK), and Dr. Maria Francia (Instituto Pasteur de Montevideo, Uruguay) for their helpful comments on the current manuscript.

## Author contributions

Conceptualisation: Z.D.V. and A.T. Main experimentation: Z.D.V. and L.R.B. Formal analysis: Z.D.V. Statistical analyses: I.C. Resources and funding: A.T. and C.H. Data Curation: Z.D.V. Writing original draft: Z.D.V. Review & Editing: Z.D.V., L.R.B., I.C., C.H., and A.T.

## Funding

## Competing interests

The authors declare no competing interests.
