## [Transparent Peer Review file · Communications Biology]

Toxoplasma gondii infection induces early host cell cycle arrest and DNA damage in primary human host cells by a MYR1-dependent mechanism.

Corresponding Author: Dr Zahady Velasquez

Version 0:

Reviewer comments:

Reviewer #1

(Remarks to the Author)

The manuscript describes the potential of *Toxoplasma gondii* infection to affect host cell genome integrity, during the early stage (first 3 hours) in primary human umbilical vein endothelial cells. However, some important points need to be addressed as below.

1. Line 340, "MOI of 1:1 or 1:0.5 (cells: parasites)" , usually we write it in an opposite way "MOI, parasites: cells"
2. Fig 1B, how many cells were counted? how many of them are positive with *T. gondii*? why G1+G2+S phase cells is over 100%?
3. Fig 2, how many cells were counted in each panel?
Fig2D , Binucleated cell , please confirm almost entire nucleus is full of γ H2AX-based DNA damage foci. (Such amount of signal is much more than the Fig 3A.) Micronucleus cell, forget to label " γ H2AX" in up-left corner.
4. What is the threshold for judge γ H2AX positive cells?
In the uninfected cells, how is the γ H2AX stained cells in S-phase, as well as in G1 and G2 phases?
5. Fig6A. What is WOS? NAC control is absent?
Fig6D. How do you explain that cells not infected with *T gondii* also have high γ H2AX signal?

Reviewer #2

(Remarks to the Author)

Summary:

The authors of this manuscript aim to investigate the early stages of *Toxoplasma* infection and how this relates to cell cycle arrest and DNA damage. They use primary HUVEC cell lines from different donors throughout the experiments, along with a low MOI of Type 1 tachyzoites and very early time points (sub 3 hours post infection). They find that the parasites induced S-phase arrest, which has been shown in previous studies, and that this is accompanied by double stranded DNA damage. The authors go on to investigate whether this damage and arrest is MYR1 and HCE1 dependent and show that HCE1 plays little to no part in the early stages of these process, but MYR1 does, suggesting another effector protein might be involved. In the second half of the manuscript, the authors investigate the downstream effects of the DNA damage, including chromosome instability and the activation of the ATM DNA repair pathway, as well as investigating whether the DNA damage is caused by ROS, as previously suggested in the literature. They conclude that early DNA damage is not ROS dependent and although the ATM DNA repair pathway is activated, it does not appear to be successful.

Conclusion:

The novelty and influence this paper will have in the field heavily relies on the experimental design. Many conclusions in the field are based on lab-specific scenarios, like a high MOI or immortalised/cancer cell lines. I applaud the authors care on ensuring their data is somewhat representative of a “truer” infection. They show that lab-specific scenarios clearly have impacts on the conclusions drawn, e.g. ROS might not be the driving force for initial DNA damage in Toxoplasma infection, contradictory to the conclusions of Zhaung et al. (2020) which might be down to experimental design, as highlighted in the discussion. The manuscript also opens questions that remain unanswered (to the best of my knowledge) in the field, like what comes first, cell cycle arrest, or DNA damage? Why would the parasite want to arrest the cells? What MYR1-dependent and non-MYR1 dependent effectors play a role in these phenotypes?

I think the manuscript, with a bit more discussion and justification as outlined below, would be complementary and beneficial to the community.

Comments of Significance:

1. I believe the ROS section could do with refinement of the text. Although it is clear and logical why testing ROS levels was done, how the experiments were done, what was used and why is very unclear. As a result, understanding the figures and context of the data becomes confusing to a non-expert in reactive oxygen species.

2. Line 86-87 states: “This finding also indicated that S-phase arrest does not exclusively depend on the presence of the parasite but might be induced by parasite secretory/excretory proteins.” In my opinion, I find this statement weak/not supported by the data and justification for the follow up use of MYR1/HCE1 mutants lacking:

- Authors state that because 64% of the cells at S-phase are infected, then the other 36% arrested cells are not infected because of exported Toxoplasma proteins. However, ~36% of uninfected cells in S-phase, of the total percentage (~35%) of cells in S-phase in infected samples, equals ~12% which could be within statistical variance of the 9.9% of cells that are in S phase in uninfected samples e.g. the percentage of uninfected cells in S phase might remain stable throughout. Reading between the lines, the authors might be referring to rhopty protein injection, where tachyzoites inject rhopty proteins into the host cell but do not completed invasion, leaving a population of uninfected bystander cells with Toxoplasma effector proteins inside (reference: Koshy et al. 2012 Plos Path). However, this is never discussed in the manuscript. Additionally, from the current literature, these proteins are exclusively rhopty proteins, and there is no data to suggest MYR1 is involved in rhopty secretion.

- Although I believe there is justification for following up with KO parasites lines with MYR1 and HCE1 because of my knowledge in the field, this is not well justified in the manuscript. Another sentence or two between lines 87 and 88 stating the reasoning with respect to the literature would go a long way to help the readers understand why the next experiments were taken. The current statement (highlighted above) is in my opinion not enough.

- Similarly, there has been some suggestions that GRA16 was involved in cell cycle and DNA damage (considering P53 modulation), but this was not an effector chosen for study. Justification for this would be helpful, whether in the introduction or in the discussion.

3. Figure 2 – no quantitation of the multipolar spindle was provided yet the conclusion that this is changed upon infection is used throughout the manuscript. Additionally, it’s very hard (dear I say almost impossible) for an untrained eye to see what is referred to as the multipolar spindle? There is no marker for the spindle poles other than arrows pointing to an area where not much is visible. I would find better presentative images of this or re-stain and image with a spindle marker rather than b-Catenin. If this cannot be done, I would consider removing or heavily minimising the conclusions around the multipolar spindle poles.

4. The author’s data suggests DNA damage happens very early on and in a MYR1-dependent fashion, along with S-phase arrest. However, the authors do not appear to discuss or speculate why the parasite might induce DNA damage? What is the benefit to the pathogen for its survival? Previous theories have suggested that cell cycle arrest could enable Toxoplasma to complete its lytic cycle before the host cell divides. What comes first: does DNA damage cause the cell to arrest in S-phase and the parasite requires this for their survival? OR is DNA damage a consequence of S-phase arrest which is the factor dependent on effector protein export? Although the data presented by the authors cannot fully answer these questions, the questions non-the-less could be included in the discussion for future gaps that could be answered.

General Comments:

Line 42-45. Has the GRA16 reference, but there is no a reference for the statement that MYR1 indicating it is part of multiprotein complex as suggested in the sentence.

Add a line showing from where the comet tails were measured in 2A would help understand how the data was produced.

It would be helpful to include a sentence in the figure 5 legend that states vinculin is the loading control.

ROS is an abbreviation, please spell out what ROS stands for the first time it is used.

Fig 6A, please clarify in the text/figure legend what WOS is (is it a dye that binds to ROS?) and what it represents in the figure (-ve control for ROS?). Additionally, as 6B is the quantification of 6A, please label them as either 6Ai/6Aii or as one sub figure under 6A. This makes it clearer to the reader that the data is from the same experiment.

The units of ROS measure (absolute count? intensity?) are missing for 6B and C.

Figure 6C, in the figure legend it would be helpful to include the dye/fluorescence marker used to determine the values.

Is the decrease in intracellular ROS of Tg infected cells dependent on the presence of MYR1? It is known that ROS are often used as an anti-infective defence mechanism in many intracellular pathogens (including Toxoplasma in macrophages), so it is interesting it goes down upon infection here. Maybe MYR1 exported or ROP proteins (which has been shown by Kochanowsky et al. 2020 (mBio) to be the case for Type III strains) help control the levels of ROS for survival.

Figure 6D, NAC+Tg images have barely any invaded tachyzoites in the presentative image. It is hard to evaluate the differences where there are no parasite infected cells.

Figure 6D, why do almost all the bystander cells have γ H2Ax expression when they are not infected? This contradicts the authors previous data in figures 1, 3 and 4.

Statistical comments:

Figures 2, 3 and 5 show the individual samples points on the bar graphs in addition to the error bars. This is very good practise and helps the reader critical analyse the data and statistical output. Please also do this for Figures 1, 4, and 6.

I noticed all the data is tested using non-parametric tests, can the authors please include in methodology whether they did a test for non-normalcy prior to this decision and what this test was.

In the methodology the authors state the data shown is in mean \pm SD, however, in figure 1 legend its states median \pm SD. Please clarify which is shown. I am not an expert in statistics, however from my understanding the statistical tests used are rank based, which should suggest the median would be represented. If the median is represented, I believe SD is not the accurate variance to show, as SD is based on normal (e.g. mean) data. I would discuss the most accurate way to present this with a more experienced statistician. The rest of the figure legends say mean \pm SD, but once again, I question if this is the most accurate way to represent the data if non-parametric tests were used.

In some of the figures, stars are shown to represent statistical significance, however, no key is provided in the legends about what the stars mean. Other times the p values are given directly on the figure. Authors should pick one for the entirety of the manuscript. I personally prefer the p-values are given directly on the figures.

Authors should double check the CDK2 WB p-value is correct. At $p=0.053$ it would technically be non-significant, other for the sample distributions shown in the figure that value seems high. If 0.053 is correct, the authors should consider rewording their language and conclusions regarding ATM vs ATR.

Figure 6 legend states a "non-parametric t-test" was used, please change this to correctly state which test was used by name like the other legends.

Version 1:

Reviewer comments:

Reviewer #1

(Remarks to the Author)

new question 1, Although the corrections were done in but not in "22878_1_merged_1721893270.pdf"
Additionally, if "MOI 1:0.5" in Fig1-6 legends are correct?

new question 2, Although some descriptions were added, the belows are misleading: "Thus, DNA-damaged cells in G1 were PCNA-/Geminin-/ γ H2Ax+, in S-phase were PCNA+/ γ H2Ax+, and in G2-phase were Geminin+/ γ H2Ax+", since no PCNA and Geminin signals from two inepanding experiment, there is no PCNA/Geminin co-labeling data.
Again, no correction was done in "22878_1_merged_1721893270.pdf"

new question 6, Sorry, I do not see the answer? Probably is in question 4.

"Nevertheless, a baseline percentage of DNA damage-positive cells is always expected." "What was surprising is that G2-phase cells showed higher DNA damage in non-infected conditions."

As the Fig6D (T. gondii pannel) shows, the top-left cell is infected and γ H2AX positive, the top-right cell is uninfected and γ H2AX positive, the bottom-left cell is uninfected and γ H2AX positive, the bottom-middle cell is uninfected and γ H2AX positive. While the rest three (center) are γ H2AX negative, and probably also the bottom-right cell. A much higher number of γ H2AX positive in T. gondii-free host cells, but only in T. gondii group, maybe these host cells also suffered T. gondii invasion, but T. gondii-left because of over-activation of respondance.

My concern is whether such DNA damage in host cells is benific for T. gondii. Possibly T. gondii apply myr1 and GRAs to induce DNA damage to block the host cell cycle, but too much DNA damages may lead to host cell death/apoptosis, which is also harmful to T. gondii, there are T. gondii.

Therefore, I suggest to quantify the γ H2AX signal intensity in each individual host cells, and also to record the possible cell fate of them. The first experiment should be simple as just looking through the picture again and to quantify the γ H2AX signal intensity together with the presence of T. gondii signal (whole or parts).

Reviewer #2

(Remarks to the Author)

Please see the attached comments on the revised copy of the paper. Thank you for making a concerted effort on updating the manuscript.

I believe the most pressing update is the statistical representation. Unfortunately, I disagree with your comments to keep it as mean \pm SD because that's what people are used to. You cannot show mean \pm SD when the test you did was a non-parametric test. This is miss-leading. How we do our statistics should be accurate and presentative of the test/statistical analysis used, not what others assume it is or what is conventional. This is why we write what we have done in the methods and figure legends. Please amend to show median with QR- a box and whiskers plot with individual data points overlayed is a good place to start, or a violin plot.

Thank you for amending the graphs to include the p-values. I believe 4B has asterisks and no value of the stats in the figure legends? Please update this one as well.

Thank you for being honest about the lack of multipolar spindles shown, and for making the changes. Just note that I believe a sentence still remains about the multipolar spindles (lines 140-141) and along with a figure reference for a figure that doesn't exist anymore. Please just double check these are deleted. You might also want to remove it from your schematic, but this is up to you.

Version 2:

Reviewer comments:

Reviewer #1

(Remarks to the Author)

on the old question "How do you explain that cells not infected with T gondii also have high γ H2AX signal?"

The authors reply by adding Fig S1B (named as Fig S2B in point-to-point response), which indicates that in the infected monolayers with a γ H2AX signal, there are significantly more T. gondii positive cells than T. gondii negative cells.

Therefore, Fig. 6D could be potentially misleading. Please replace it with another more representative photo.

Reviewer #2

(Remarks to the Author)

Thank for amending the manuscript, the violin plots look fantastic!

One final update: The Statistical Methods currently states: "All data were expressed as mean \pm SD from at least three independent experiments." and will need to be updated to say median is shown to reflect the changes you've made.

Reviewer #1 (Remarks to the Author):

The manuscript describes the potential of *Toxoplasma gondii* infection to affect host cell genome integrity, during the early stage (first 3 hours) in primary human umbilical vein endothelial cells. However, some important points need to be addressed as below.

We appreciate your constructive comments on our manuscript. Following your suggestions, we have added the required information and hope that we have addressed your concerns.

1. Line 340, "MOI of 1:1 or 1:0.5 (cells: parasites)", usually we write it in an opposite way "MOI, parasites: cells"

We have amended the sentence as you suggested. Please, refer to line 359.

2. Fig 1B, how many cells were counted? how many of them are positive with *T. gondii*? why G1+G2+S phase cells is over 100%?

Thank you for highlighting the confusion regarding Fig 1 quantifications. We noticed that we did not specify the total number of counted cells, only the number of biological donors used. This information can now be found in Supplementary Figure S1A, where we detail the total number of cells counted at each time point and for each strain. Additionally, this information has been added to the manuscript text.

Regarding the methods used to count the cells in each cell cycle phase, we apologize for the lack of clarity in our explanation on lines 377-382. After testing several offers of antibodies to detect PCNA and Geminin isotypes, the same isotypes of antibodies were used for PCNA and Geminin detection, making colocalization impossible. Consequently, cell cycle phases were quantified in two independent staining assays. The S-phase was calculated in cells stained with PCNA and DAPI. The total number of PCNA-positive cells was assigned as S-phase cells. For the G1 and G2 phases, cells were stained with Geminin and DAPI. Geminin-positive cells were assigned to the G2-phase, and Geminin-negative cells to the G1-phase. Therefore, the sum of all three phases did not always total 100%. For example, one image had 51 cells in total, of which 20 were PCNA-positive (S-phase), resulting in 39.2% S-phase cells. Conversely, the total number of Geminin-stained cells was 69, with 41 being Geminin-positive (G2-phase) and 28 Geminin-negative (G1-phase), equating to 59.4% G2-phase and 40.6% G1-phase cells. The inability to perform the two staining together led us to present the data in separate graphs.

We understand the importance of clarifying the scoring criteria for our microscopy-based assay in identifying cell cycle phases. We have now provided a more detailed description in the revised materials and methods section to address this concern. Please refer to lines 396-402.

3. Fig 2, how many cells were counted in each panel?

On Average, 1400 cells were counted in each experimental condition shown in Fig 2A, B and C. This information is now included in the legend caption.

Fig2D, Binucleated cell, please confirm almost entire nucleus is full of γ H2AX-based DNA damage foci. (Such amount of signal is much more than the Fig 3A) Micronucleus cell, forget to label " γ H2AX" in up-left corner.

I appreciate your comments on binucleated cell DNA damage, as this was an observation that also surprised us. Indeed, the nuclei in non-binucleated cells display a smaller number of DNA damage foci

compared to binucleated cells, with the only exception being the HCE1 strain (Fig 3A). Initially, the intensity of the DNA damage foci signal appeared higher in binucleated cells, leading us to suspect a staining issue. Therefore, we repeated the immunofluorescence assay (IFA), but the results were consistent, suggesting that binucleated cells indeed exhibit higher DNA damage than mononucleated cells.

About the micronuclei picture information, we have corrected the mistake.

4. What is the threshold for judge γ H2AX positive cells?

Positive γ H2AX cells were those with a fluorescence mean of 400, below this number, the cells were not added to the counting. Also, we selected those with a clear dot shape of the staining.

In the uninfected cells, how is the γ H2AX stained cells in S-phase, as well as in G1 and G2 phases?

I apologize, but I don't fully understand your question. Therefore, I will provide an explanation based on what I believe you asked. Also, I will use the explanation to answer your last question of the revision.

In cell culture conditions, we recognize that cells quickly display DNA damage foci when they experience changes in environmental conditions such as prolonged time outside the incubator, abrupt temperature reductions, or aggressive pipetting during the trypsinization process. Since we work exclusively with primary cells, this can be explained by their unfamiliarity with a 2D structure, where their only contact is with the plastic of the flask and the air above. Also, this could be due to the highly restricted medium that scientists use in the cell culture routine in which, for instance, there is any associated hormonal regulation. Therefore, we have trained ourselves to avoid these perturbations during cell culture. Nevertheless, a baseline percentage of DNA damage-positive cells is always expected. Taking into account this, and that we wanted to correlate non-infected (n.i.) and *T. gondii*-infected conditions, we conducted all experimental conditions in all cell donors in parallel. When counting DNA damage-positive cells, we only included those positive for γ H2AX, normalized by the number of DAPI-positive cells in the same field of view. Consequently, figure 4B only shows those cells, in the monolayer, that were positive for DNA damage. What was surprising is that G2-phase cells showed higher DNA damage in non-infected conditions. However, we do not have a definitive explanation for this. One possibility is that G2-phase cells, having just completed DNA synthesis, maybe in the process of correcting DNA damage through intracellular repair pathways. Studies on cell cycle control have shown that DNA damage checkpoints in the G1 and G2 phases work to repair any damage produced in previous stages, sometimes arresting the cell to allow time for repair (DOI:<https://doi.org/10.1016/j.cels.2017.09.015>).

5. Fig6A. What is WOS? NAC control is absent?

We apologize for not providing an explanation of "WOS" in the figure caption. WOS stands for "without staining." This definition has now been added to the figure caption.

Regarding NAC, this information is indeed present in the figure. We did not include it in the histogram to focus attention on the effects of *T. gondii* infection and NAC treatment on infected cells. However, you can see the NAC-alone values incorporated in the graph in Figure 6B, third column.

Fig6D. How do you explain that cells not infected with *T gondii* also have high γ H2AX signal?

This question was answered in the question number.

Reviewer #2 (Remarks to the Author):

Summary:

The authors of this manuscript aim to investigate the early stages of *Toxoplasma* infection and how this relates to cell cycle arrest and DNA damage. They use primary HUVEC cell lines from different donors throughout the experiments, along with a low MOI of Type 1 tachyzoites and very early time points (sub 3 hours post infection). They find that the parasites induced S-phase arrest, which has been shown in previous studies, and that this is accompanied by double stranded DNA damage. The authors go on to investigate whether this damage and arrest is MYR1 and HCE1 dependent and show that HCE1 plays little to no part in the early stages of these process, but MYR1 does, suggesting another effector protein might be involved. In the second half of the manuscript, the authors investigate the downstream effects of the DNA damage, including chromosome instability and the activation of the ATM DNA repair pathway, as well as investigating whether the DNA damage is cause by ROS, as previously suggested in the literature. They conclude that early DNA damage is not ROS dependent and although the ATM DNA repair pathway is activated, it does not appear to be successful.

Conclusion:

The novelty and influence this paper will have in the field heavily relies on the experimental design. Many conclusions in the field are based on lab-specific scenarios, like a high MOI or immortalised/cancer cell lines. I applaud the authors care on ensuring their data is somewhat representative of a “truer” infection. They show that lab-specific scenarios clearly have impacts on the conclusions drawn, e.g. ROS might not be the driving force for initial DNA damage in *Toxoplasma* infection, contradictory to the conclusions of Zhaung et al. (2020) which might be down to experimental design, as highlighted in the discussion. The manuscript also opens questions that remain unanswered (to the best of my knowledge) in the field, like what comes first, cell cycle arrest, or DNA damage? Why would the parasite want to arrest the cells? What MYR1-dependent and non-MYR1 dependent effectors play a role in these phenotypes?

I think the manuscript, with a bit more discussion and justification as outlined below, would be complementary and beneficial to the community.

Thank you for your valuable and insightful revision of our work. We deeply appreciate your recognition of the novelty and potential impact of our experimental design on the field. Your comments about our efforts to ensure that our data is representative of a "truer" infection scenario using only primary cells is particularly gratifying. While we value and respect what our worldwide colleagues have published, we agree with you that the data on ROS modulation by *T. gondii* in HeLa cells are unlikely something observed in tissues or animals. This was the exact reason why we studied whether our primary cells also modulated ROS after infection.

Your suggestions for further discussion and justification are invaluable, and we are committed to incorporating them to enhance the manuscript's contribution to the scientific community.

Thank you once again for your thoughtful and constructive feedback. We hope that we have addressed your concerns in the responses provided below.

Comments of Significance:

1. I believe the ROS section could do with refinement of the text. Although it is clear and logical why testing ROS levels was done, how the experiments were done, what was used and why is very unclear. As a result, understanding the figures and context of the data becomes confusing to a non-expert in reactive oxygen species.

Thanks for highlighting that the experiments and results regarding ROS were not clearly explained. We reviewed the text accordingly and added new sentences to make this information clearer to readers.

2. Line 86-87 states: “This finding also indicated that S-phase arrest does not exclusively depend on the presence of the parasite but might be induced by parasite secretory/excretory proteins.” In my opinion, I find this statement weak/not supported by the data and justification for the follow up use of MYR1/HCE1 mutants lacking:

- Authors state that because 64% of the cells at S-phase are infected, then the other 36% arrested cells are not infected because of exported Toxoplasma proteins. However, ~36% of uninfected cells in S-phase, of the total percentage (~35%) of cells in S-phase in infected samples, equals ~12% which could be within statistical variance of the 9.9% of cells that are at in S phase in uninfected samples e.g. the percentage of uninfected cells in S phase might remain stable throughout. Reading between the lines, the authors might be referring to rhoptry protein injection, where tachyzoites inject rhoptry proteins into the host cell but do not completed invasion, leaving a population of uninfected bystander cells with Toxoplasma effector proteins inside (reference: Koshy et al. 2012 Plos Path). However, this is never discussed in the manuscript. Additionally, from the current literature, these proteins are exclusively rhoptry proteins, and there is no data to suggest MYR1 is involved in rhoptry secretion.

We are glad you brought this point to our attention during the revision because we were not aware of how it sounded when isolated from the whole text. First, I apologize for this confusing conclusion regarding our cell cycle results. Secondly, I agree with you that this statement does not fit with our results, and we have erased it from the text.

- Although I believe there is justification for following up with KO parasites lines with MYR1 and HCE1 because of my knowledge in the field, this is not well justified in the manuscript. Another sentence or two between lines 87 and 88 stating the reasoning with respect to the literature would go a long way to help the readers understand why the next experiments were taken. The current statement (highlighted above) is in my opinion not enough.

- Similarly, there has been some suggestions that GRA16 was involved in cell cycle and DNA damage (considering P53 modulation), but this was not an effector chosen for study. Justification for this would be helpful, whether in the introduction or in the discussion.

Following your recommendation, we added the information that led us to use the MYR1 and HCE1 mutants in our study. We hope that the results section now provides a better understanding of the results shown and the research pipeline. Please refer to lines 86-96.

3. Figure 2 – no quantitation of the multipolar spindle was provided yet the conclusion that this is changed upon infection is used throughout the manuscript. Additionally, it’s very hard (dear I say almost impossible) for an untrained eye to see what is referred to as the multipolar spindle? There is no marker for the spindle poles other than arrows pointing to an area where not much is visible. I would find better presentative images of this or re-stain and image with a spindle marker rather than b-

Catenin. If this cannot be done, I would consider removing or heavily minimising the conclusions around the multipolar spindle poles.

Your observation about what we have presented in Figure 2 is completely right. Since there are a couple of published papers showing that *T. gondii* induces multipolar spindle in several cell types, we allowed ourselves the freedom to only mention that we also observed this phenomenon only after 3 hpi. Published was shown at 24 hpi only. However, this was only an observation and following your indications, we have erased the image with the chromosome segregation problem as well as our conclusions about this observation.

4. The author's data suggests DNA damage happens very early on and in a MYR1-dependent fashion, along with S-phase arrest. However, the authors do not appear to discuss or speculate why the parasite might induce DNA damage? What is the benefit to the pathogen for its survival? Previous theories have suggested that cell cycle arrest could enable Toxoplasma to complete its lytic cycle before the host cell divides. What comes first: does DNA damage cause the cell to arrest in S-phase and the parasite requires this for their survival? OR is DNA damage a consequence of S-phase arrest which is the factor dependent on effector protein export? Although the data presented by the authors cannot fully answer these questions, the questions non-the-less could be included in the discussion for future gaps that could be answered.

We appreciate your insights regarding the correlation between our results and the first stages of infection. We agree that we did not elaborate extensively on our hypothesis linking cell cycle arrest and DNA damage induction. This is because we consider that our results highlight the need to investigate the first minutes of the infection and the potential role of MYR1 in this process. However, the limited scope of our current manuscript makes it not appropriate to hypothesize extensively with only the hint that MYR1 might have a potential role in the host cell cycle arrest and the DNA damage induction.

Our study was initially designed to focus on the first 3h p.i but we were surprised by the earlier effect on cell cycle and DNA damage. Therefore, we decided to leave this topic for further manuscripts, which we are currently working on. Nevertheless, we would take your advice and acknowledge these data as a future research gap in our discussion. We will incorporate some of your comments, as they align with our perspectives. Thank you for your valuable feedback.

General Comments:

Line 42-45. Has the GRA16 reference, but there is no a reference for the statement that MYR1 indicating it is part of multiprotein complex as suggested in the sentence.

Per your suggestion, we have now included the necessary citations.

Add a line showing from where the comet tails were measured in 2A would help understand how the data was produced.

The modification was done in lines 115-123.

It would be helpful to include a sentence in the Figure 5 legend that states vinculin is the loading control.

Thanks for noticing the absence of a description of what was the vinculin's role in the WB. The information was added to the legend.

ROS is an abbreviation, please spell out what ROS stands for the first time it is used.

Done.

Fig 6A, please clarify in the text/figure legend what WOS is (is it a dye that binds to ROS?) and what it represents in the figure (-ve control for ROS?). Additionally, as 6B is the quantification of 6A, please label them as either 6Ai/6Aii or as one sub figure under 6A. This makes it clearer to the reader that the data is from the same experiment.

Thanks for highlighting the missing information about WOS and the confusion between figures 6A and B. Following your recommendations, we fused 6A and B in only Figure 6A. Also, the definition of WOS (without staining) is now stated in the figure legend and the result text.

The units of ROS measure (absolute count? intensity?) are missing for 6B and C.

We apologize for this confusion; intracellular ROS was measured using the DCFH-DA probe and the values are informed as the mean of fluorescence intensity (MFI) whilst extracellular ROS was detected by using Amplex red staining and the units informed in the graph are relative fluorescence intensity (RFI). Both are written in the graph axes, but we added them now also in the figure legend.

Figure 6C, in the figure legend it would be helpful to include the dye/fluorescence marker used to determine the values.

Done.

Is the decrease in intracellular ROS of Tg infected cells dependent on the presence of MYR1? It is known that ROS are often used as an anti-infective defence mechanism in many intracellular pathogens (including *Toxoplasma* in macrophages), so it is interesting it goes down upon infection here. Maybe MYR1 exported or ROP proteins (which has been shown by Kochanowsky et al. 2020 (mBio) to be the case for Type III strains) help control the levels of ROS for survival.

We appreciate your interest in our results. Initially, we quantified ROS in all three strains using luminometry. This involved incubating cells with luminol and isoluminol. As shown in the graphs below, we did not observe any significant changes in extra- or intracellular ROS levels in MYR1- or HCE1-deficient strains compared to the KU80 strain.

Despite these results indicating no modulation of ROS, previous studies have shown that ROS levels increase in HeLa-infected cells. Therefore, we proceeded with a more sensitive method: FACS analysis of live cells incubated with the DCFH-DA probe. Since no differences were observed across all strains, we focused our analysis on TgRH, as detailed in the manuscript. Thus, the mutant strains were not analyzed with DCFH-DA by FACS, given that the luminol/isoluminol results suggest that none of the *T. gondii* strains increases intracellular or extracellular ROS.

Regarding your comments on macrophages and other strains, we agree with your hypothesis. We have been working with human and bovine neutrophils as well as macrophages confronted/infected with *Toxoplasma*, but we did not see a positive or negative impact on ROS, even though a recent study described an increase in extracellular ROS in human PMNs (<https://doi.org/10.3389/fimmu.2023.1282278>). However, none of us have tried any mutant strains to assess their effect on innate immunity. We will keep your comments in mind for future experiments to test whether the MYR1 mutant could be important in the immune reaction against infection.

Figure 6D, NAC+Tg images have barely any invaded tachyzoites in the presentative image. It is hard to evaluate the differences where there are no parasite infected cells.

Thanks for highlighting the differences in the infection shown in Figure 6D. We took new pictures showing similar infections in the field of view and these pictures are in the new version of Figure 6D.

Figure 6D, why do almost all the bystander cells have γ H2Ax expression when they are not infected? This contradicts the authors previous data in figures 1, 3 and 4.

I apologize, but I don't fully understand your question. Therefore, I will provide an explanation based on what I believe you asked. First of all, in Figure 1 there is no quantification about DNA damage foci (γ H2Ax), but maybe this was only a typo problem. Thus, I will refer to only Figures 3 and 4. When we described the DNA damage by the use of γ H2Ax staining, we only showed quantification from the whole n.i. and infected monolayer, therefore, we have no data about the infected cells and bystander cells regarding the DNA damage. Our comparisons are only referred to the non-infected monolayer as an independent experiment and never splitting data from the same infected monolayer (Tg vs bystander or n.i. cells). As we also explained to Reviewer 1, in cell culture conditions, we recognize that cells quickly display DNA damage foci when they experience changes in environmental conditions such as prolonged time outside the incubator, abrupt temperature reductions, or aggressive pipetting during the trypsinization process. Since we work exclusively with primary cells, this can be explained by their unfamiliarity with a 2D structure, where their only contact is with the plastic of the flask and the air above. Also, this could be due to the highly restricted medium that scientists use in the cell culture routine in which, for instance, there is any associated hormonal regulation. Therefore, we have trained ourselves to avoid these perturbations during cell culture. Nevertheless, a baseline percentage of DNA damage in foci-positive cells is always expected. Considering this fact and that our experimental design sought to correlate non-infected (n.i.) and *T. gondii*-infected conditions, we conducted all experimental conditions in all cell donors in parallel.

When counting DNA damage-positive cells, we only included those positive for γ H2AX, normalized by the number of DAPI-positive cells in the same field of view. Consequently, figure 4B only shows those cells, in the monolayer, that were positive for DNA damage. What was surprising is that G2-phase cells showed higher DNA damage in non-infected conditions. However, we do not have a definitive explanation for this. One possibility is that G2-phase cells, having just completed DNA synthesis, maybe in the process of correcting DNA damage through intracellular repair pathways. Studies on cell cycle control have shown that DNA damage checkpoints in the G1 and G2 phases work to repair any damage

produced in previous stages, sometimes arresting the cell to allow time for repair (DOI:<https://doi.org/10.1016/j.cels.2017.09.015>).

Statistical comments:

Figures 2, 3 and 5 show the individual samples points on the bar graphs in addition to the error bars. This is very good practise and helps the reader critical analyse the data and statistical output. Please also do this for Figures 1, 4, and 6.

All graphs were amended with bars and points. The only exception was Figure 1B where adding the p-value makes the graph hard to interpret. However, the value of p was added to the Figure legend.

I noticed all the data is tested using non-parametric tests, can the authors please include in methodology whether they did a test for non-normalcy prior to this decision and what this test was.

In the methodology the authors state the data shown is in mean +/- SD, however, in figure 1 legend its states median +/- SD. Please clarify which is shown. I am not an expert in statistics, however from my understanding the statistical tests used are rank based, which should suggest the median would be represented. If the median is represented, I believe SD is not the accurate variance to show, as SD is based on normal (e.g. mean) data. I would discuss the most accurate way to present this with a more experienced statistician. The rest of the figure legends say mean +/- SD, but once again, I question if this is the most accurate way to represent the data if non-parametric tests were used.

Thanks for this comment because it was something that we have found several times in our research statistical analysis. The chosen tests were based on the assumptions of the tests and the distribution of the data. Regarding the results calculated as % of cells with X effect, since the data is already normalized (%) and thus continuous, the conventional recommendation is to run non-parametric tests. In this case, the selection of the test was not based on the result of a normality test but on the type of data (i.e. percentages). In the case of discrete data (i.e. long of the tails), some datasets did not pass the normality test, based on the results of D'Agostino & Pearson and Shapiro-Wilk normality tests. This information is now added in the material and Methods section.

We completely agree with you that the median is the correct statistical parameter to be reported on the plots when non-parametric tests are used, as correctly pointed out by the reviewer since are based on median comparisons and then ranks. However, I think the readers (and sometimes reviewers, editors etc) are more used to assuming that the mean \pm SD is represented in the figures, even though the median is more appropriate. Hopefully, the reviewer agrees to maintain mean \pm SD as the way to show the graphs, just because is more conventional.

In some of the figures, stars are shown to represent statistical significance, however, no key is provided in the legends about what the stars mean. Other times the p values are given directly on the figure. Authors should pick one for the entirety of the manuscript. I personally prefer the p-values are given directly on the figures.

Thanks for noticing this discrepancy. Following your recommendation we changed all asterisks by p-value in graphs to show directly whether they were or not statistically significant. The only exception is Figure 1B in which we kept the asterisks to avoid an overcrowded graph, but we informed the p-value of each statistic written in the graph.

Authors should double check the CDK2 WB p-value is correct. At $p=0.053$ it would technically be non-significant, other for the sample distributions shown in the figure that value seems high. If 0.053 is correct, the authors should consider rewording their language and conclusions regarding ATM vs ATR.

You are correct about CDK2 statistical significance. We are aware that they were non-significant, and we kept it only because we wanted to show that even though CDK2 expression was evident lower, they were not statistically significant. However, we noticed with your question that it leads to misunderstanding, and we erased the p-value.

Figure 6 legend states a “non-parametric t-test” was used, please change this to correctly state which test was used by name like the other legends.

All figure's legends were changed indicating the right statistical test used.

Reviewer #2 (Remarks to the Author):

We would like to thank your thoughtful revision of our manuscript. We believe that now it is a much better and clearer version of it.

As you suggested, we have changed the plots to violin showing the medium and overlapping the individual points. We hope that now the manuscript is suitable for publication.

3. Figure 2 – no quantitation of the multipolar spindle was provided yet the conclusion that this is changed upon infection is used throughout the manuscript. Additionally, it's very hard (dear I say almost impossible) for an untrained eye to see what is referred to as the multipolar spindle? There is no marker for the spindle poles other than arrows pointing to an area where not much is visible. I would find better presentative images of this or re-stain and image with a spindle marker rather than b-Catenin. If this cannot be done, I would consider removing or heavily minimising the conclusions around the multipolar spindle poles.

Your observation about what we have presented in Figure 2 is completely right. Since there are a couple of published papers showing that *T. gondii* induces multipolar spindle in several cell types, we allowed ourselves the freedom to only mention that we also observed this phenomenon only after 3 hpi. Published was shown at 24 hpi only. However, this was only an observation and following your indications, we have erased the image with the chromosome segregation problem as well as our conclusions about this observation.

Reviewer: Thank you for being honest about this, and for updating. Just note that I believe a sentence still remains about the multipolar spindles (lines 140-141) and along with a figure reference for a figure that doesn't exist anymore. Please just double check these are deleted. You might also want to remove it from your schematic, this is up to you.

Thanks for highlighting that the sentence remains in the text. We erased it now and we removed mitotic spindle draw from scheme 1 as well.

Statistical comments:

Figures 2, 3 and 5 show the individual samples points on the bar graphs in addition to the error bars. This is very good practise and helps the reader critical analyse the data and statistical output. Please also do this for Figures 1, 4, and 6.

All graphs were amended with bars and points. The only exception was Figure 1B where adding the p-value makes the graph hard to interpret. However, the value of p was added to the Figure legend.

Reviewer: Thank you for amending the graphs. I believe 4B has asterisks and no value of the stats in the figure legends? Please update this one as well.

You are completely right. Now we change the graph by adding violin graphs and the p-value of each experimental condition.

I noticed all the data is tested using non-parametric tests, can the authors please include in methodology whether they did a test for non-normality prior to this decision and what this test was. In the methodology the authors state the data shown is in mean +/- SD, however, in figure 1 legend its states median +/- SD. Please clarify which is shown. I am not an expert in statistics, however from my understanding the statistical tests used are rank based, which should suggest the median would be represented. If the median is represented, I believe SD is not the accurate variance to show, as SD is based on normal (e.g. mean) data. I would discuss the most accurate way to present this with a more experienced statistician. The rest of the figure legends say mean +/- SD, but once again, I question if this is the most accurate way to represent the data if non-parametric tests were used.

We completely agree with you that the median is the correct statistical parameter to be reported on the plots when non-parametric tests are used, as correctly pointed out by the reviewer since are based on median comparisons and then ranks. However, I think the readers (and sometimes reviewers, editors etc) are more used to assuming that the mean \pm SD is represented in the figures, even though the median is more appropriate. Hopefully, the reviewer agrees to maintain mean \pm SD as the way to show the graphs, just because is more conventional.

Reviewer: Unfortunately, I disagree. You cannot show mean \pm SD when the test you did was a non-parametric test. This is miss-leading. How we do our statistics should be accurate and presentative of the correct test/statistical analysis, not what others assume it is or what is conventional. This is why we write what we have done in the methods and figure legends. Please amend to show median with QR- a box and whiskers plot with individual data points overlayed is a good place to start, or a violin plot.

Following your recommendations we have changed all graphs to violin plots, showing the median and overlapping individual points. The only exceptions are WB and ROS plots.

In some of the figures, stars are shown to represent statistical significance, however, no key is provided in the legends about what the stars mean. Other times the p values are given directly on the figure. Authors should pick one for the entirety of the manuscript. I personally prefer the p-values are given directly on the figures.

Thanks for noticing this discrepancy. Following your recommendation we changed all asterisks by p-value in graphs to show directly whether they were or not statistically significant. The only exception is Figure 1B in which we kept the asterisks to avoid an overcrowded graph, but we informed the p-value of each statistic written in the graph.

Reviewer: Thank you for amending the graphs. I believe 4B has asterisks and no value of the stats in the figure legends?

You are right. Now we added the p-value of each experimental condition.

Reviewer 1 – Rebuttal #2

New question 1, Although the corrections were done in but not in 22878_1_merged_1721893270.pdf"

We apologize for this mistake. To avoid further misunderstanding with the files, now we have highlighted the amends done on the latest version of the manuscript.

Additionally, if "MOI 1:0.5" in Fig1-6 legends are correct?

All MOI was changed as you asked to parasites: cells. Now they are highlighted to easily recognise them.

New question 2, Although some descriptions were added, the belows are misleading: "Thus, DNA-damaged cells in G1 were PCNA-/Geminin-/ γ H2Ax+, in S-phase were PCNA+/ γ H2Ax+, and in G2-phase were Geminin+/PCNA-/ γ H2Ax+", since no PCNA and Geminin signals from two inepanding experiment, there is no PCNA/Geminin co-labeling data.

Again, no correction was done in "22878_1_merged_1721893270.pdf"

Following your recommendations, the misleading sentence was amended.

New question 6, Sorry, I do not see the answer? Probably is in question 4.

"Nevertheless, a baseline percentage of DNA damage-positive cells is always expected." "What was surprising is that G2-phase cells showed higher DNA damage in non-infected conditions."

As the Fig6D (T. gondii pannel) shows, the top-left cell is infected and γ H2AX positive, the top-right cell is uninfected and γ H2AX positive, the bottom-left cell is uninfected and γ H2AX positive, the bottom-middle cell is uninfected and γ H2AX positive. While the rest three (center) are γ H2AX negative, and probably also the bottom-right cell. A much higher number of γ H2AX positive in T. gondii-free host cells, but only in T. gondii group, maybe these host cells also suffered T. gondii invasion, but T. gondii-left because of over-activation of respondence.

My concern is whether such DNA damage in host cells is benific for T. gondii. Possibly T. gondii apply myr1 and GRAs to induce DNA damage to block the host cell cycle, but too much DNA damages may lead to host cell death/apoptosis, which is also harmful to T. gondii, there are T. gondii. Therefore, I suggest to quantify the γ H2AX signal intensity in each individual host cells, and also to record the possible cell fate of them. The first experiment should be simple as just looking through the picture again and to quantify the γ H2AX signal intensity together with the presence of T. gondii signal (whole or parts).

We agree with your concern about the DNA damage foci being expressed on non-infected and T. gondii-infected cells as well. It was something that we also faced during the discussion of our results. To make the explanation clearer, I will divide your questions.

1.- "My concern is whether such DNA damage in host cells is benific for T. gondii. Possibly T. gondii apply myr1 and GRAs to induce DNA damage to block the host cell cycle, but too much DNA damages may lead to host cell death/apoptosis, which is also harmful to T. gondii, there are T. gondii"

Our findings showed that *T. gondii* infection induced DNA damage foci in both, non-infected and infected cells inside the same infected monolayer. These initial data were performed in infected cells where the parasite underwent several divisions (12 hpi), and it was in the middle of its intracellular

development. Given that *T. gondii* is known to scavenge purines from the host cells, we hypothesise that S-phase-arrest and DNA damage were a collateral effect of the nucleotide scavenger. When cells do not count with nucleotides enough, they arrest the S-phase and the DNA replication is delayed until these macromolecules are available again. To test this hypothesis, we repeat the experiment at times that the parasite was not even duplicated such as 15 min p.i and 3 h p.i. The first division of the tachyzoite takes place at about 3-6 h p.i. Therefore, these selected times ensure that tachyzoites were not still using nucleotides from the host cell in their own DNA synthesis process previously division. However, we observed DNA damage as soon as the parasite just completed the attachment and invasion of the host cell (15 min p.i.) suggesting that nucleotide depletion is probably not triggering the arrest or DNA damage. Then, we tested two recognized mutant parasites to interfere with the host cell division, HCE1 and MYR1 mutants. Surprisingly for us, only MYR1 mutant decreased DNA damage induction suggesting GRAs or downstream proteins in intracellular pathways modulations. All that I described was indeed exciting for us, but they are premature to use our finding extrapolating why *T. gondii* modulates the host cell in ways that can kill it. Our main concern now: i) are these modulations collateral damages of the infection? Or are they a strategy from the parasite to survive inside the host? Nevertheless, we are committed to answering this question and we are currently working on deciphering the role of MYR1, by treating cells with a c-Myc inhibitor. These results will shed light on whether the DNA damage and cell cycle arrest are indeed related to the host cell c-Myc activity modulation or directly by the translocation of some specific GRA proteins. Regarding apoptosis, it is something that we already observed in our culture cells. Also, it was published last year (<https://doi.org/10.1186/s13071-023-05991-y>). Following our culture observations, we are now studying when this happens or whether these cells were damaged and then triggered apoptosis or both are independent processes. I hope in 2025 we will be able to share some new data referring which is the *T. gondii* advantage when the host cell is not allowed to complete its cell cycle and its DNA is damaged.

2.- Therefore, I suggest to quantify the γ H2AX signal intensity in each individual host cells, and also to record the possible cell fate of them. The first experiment should be simple as just looking through the picture again and to quantify the γ H2AX signal intensity together with the presence of *T. gondii* signal (whole or parts).

The signal of γ H2AX was indeed measured in each cell independently. Several pictures were taken randomly and each cell in the field of view was counted to be positive or negative for γ H2AX (Figure 3). As you suggested we added in supplementary figure 2B, the quantification of γ H2AX in infected and non-infected cells in infected monolayers. In this new figure, you can observe that both, non-infected and *T.g.*-infected cells are damaged in their DNA when they are in an infected monolayer. About the live cell imaging suggestion, it is something that we have been trying for months now. We are committed to working only with primary cells which makes it difficult to transfect plasmids for expressing conjugated proteins for DNA damage detection. We have tested several methods of transfection, electroporation and special transfection complexes to incorporate plasmid inside cells without any luck. HUVEC are sensitive to changes in the environment and react immediately triggering DNA damage that potentially will be masking the parasite damage induction. Currently, we are trying to use vital staining but we have not found a suitable DNA damage marker. Therefore, I am afraid that we won't be able to offer you results in a live cell.

Reviewer 1 – Rebuttal #3

On the old question "How do you explain that cells not infected with *T. gondii* also have high γ H2AX signal?"

The authors reply by adding Fig S1B (named as Fig S2B in point-to-point response), which indicates that in the infected monolayers with a γ H2AX signal, there are significantly more *T. gondii* positive cells than *T. gondii* negative cells.

Therefore, Fig. 6D could be potentially misleading. Please replace it with another more representative photo.

I would like to base my response on Figures 3B and 6D, which I believe are relevant to the reviewer's query. DNA damage is a natural process that occurs in cells throughout their lifecycle. It may happen during the DNA synthesis phase or other phases when cells are subjected to stress. This damage is not necessarily permanent, as cells can activate various DNA repair pathways to correct the damage and continue with their cell cycle. Therefore, it is possible to detect DNA damage at different times during a cell's lifecycle due to the dynamic nature of this process, which can be triggered and repaired multiple times. Also, the intensity of the damage can be higher or lesser depending on the suffered stress.

In our study, we maintained strict conditions by working with all donors and experimental setups simultaneously. Both control and infected samples were collected at the same time for each time point, ensuring consistency across the experimental conditions. This approach ensures that the non-infected (n.i.) condition serves as the closest equivalent to the infected plate, minus the parasite. It is worth noting that experiments conducted with different donors or aimed at measuring other factors, such as ROS levels, may yield different percentages of DNA damage in control samples. Therefore, showed figure control indeed corresponds to those displayed in the graph bar.

Referring to the DNA damage percentages presented in Figures 3B and 6D, here are the relevant data:

Figure 3B

Time (h p.i.)	DNA damage (γ H2AX signal)		Delta Infection
	n.i.	T.g.	
0.25	7.3%	35.7%	28.4%
1	6.7%	37%	30.3%
3	9.8%	43%	33.2%

Figure 6D

Stimuli	DNA damage (γ H2AX signal)	
	n.i.	Delta Infection
n.i.	1.9%	-
T. g.	25.6%	23.7%
NAC + T. g.	22.2%	20.3%

When comparing both experiments, it is evident that, regardless of the assay, the difference in DNA damage percentages between non-infected and infected cells is consistent. In cases where non-infected cells exhibited a higher percentage of DNA damage, the infected conditions similarly showed an increase. We attribute this variation to the handling of the plates during the experiments. Both control and infected cells were taken out of the incubator for the infection process and subsequently returned, remaining outside for periods of about 10 min and then were collected at 15 minutes, 1 hour, or 3 hours. This handling likely disrupted the cellular environment (e.g., CO₂ levels, humidity, temperature), which may have led to some DNA damage, as indicated by the γ H2AX signal.

In conclusion, we consider a 9.8% DNA damage level not significantly high, particularly when the infection condition exhibited a 33% increase compared to the control. This outcome can be reasonably attributed to normal experimental handling. Moreover, we also demonstrate that cells triggered the DNA repair pathway suggesting that any insult is trying to be immediately repaired by the host cell.

As a conclusion, we respectfully disagree with the suggestion to modify the images in Figure 6D. The images accurately depict the control cells as well the infected (with or without NAC). Also, the quantification reflects the data collected from all tested donors which in our opinion is not potentially misleading readers. We hope that the reviewer finds our explanation reasonable and accepts our response to this concern.

Reviewer #2: Revision 3

One final update: The Statistical Methods currently states: "All data were expressed as mean \pm SD from at least three independent experiments." and will need to be updated to say median is shown to reflect the changes you've made.

We have amended it.

Reviewer #2 (Remarks to the Author):

Summary:

The authors of this manuscript aim to investigate the early stages of *Toxoplasma* infection and how this relates to cell cycle arrest and DNA damage. They use primary HUVEC cell lines from different donors throughout the experiments, along with a low MOI of Type 1 tachyzoites and very early time points (sub 3 hours post infection). They find that the parasites induced S-phase arrest, which has been shown in previous studies, and that this is accompanied by double stranded DNA damage. The authors go on to investigate whether this damage and arrest is MYR1 and HCE1 dependent and show that HCE1 plays little to no part in the early stages of these process, but MYR1 does, suggesting another effector protein might be involved. In the second half of the manuscript, the authors investigate the downstream effects of the DNA damage, including chromosome instability and the activation of the ATM DNA repair pathway, as well as investigating whether the DNA damage is cause by ROS, as previously suggested in the literature. They conclude that early DNA damage is not ROS dependent and although the ATM DNA repair pathway is activated, it does not appear to be successful.

Conclusion:

The novelty and influence this paper will have in the field heavily relies on the experimental design. Many conclusions in the field are based on lab-specific scenarios, like a high MOI or immortalised/cancer cell lines. I applaud the authors care on ensuring their data is somewhat representative of a “truer” infection. They show that lab-specific scenarios clearly have impacts on the conclusions drawn, e.g. ROS might not be the driving force for initial DNA damage in *Toxoplasma* infection, contradictory to the conclusions of Zhaung et al. (2020) which might be down to experimental design, as highlighted in the discussion. The manuscript also opens questions that remain unanswered (to the best of my knowledge) in the field, like what comes first, cell cycle arrest, or DNA damage? Why would the parasite want to arrest the cells? What MYR1-dependent and non-MYR1 dependent effectors play a role in these phenotypes?

I think the manuscript, with a bit more discussion and justification as outlined below, would be complementary and beneficial to the community.

Thank you for your valuable and insightful revision of our work. We deeply appreciate your recognition of the novelty and potential impact of our experimental design on the field. Your comments about our efforts to ensure that our data is representative of a "truer" infection scenario using only primary cells is particularly gratifying. While we value and respect what our worldwide colleagues have published, we agree with you that the data on ROS modulation by *T. gondii* in HeLa cells are unlikely something observed in tissues or animals. This was the exact reason why we studied whether our primary cells also modulated ROS after infection.

Your suggestions for further discussion and justification are invaluable, and we are committed to incorporating them to enhance the manuscript's contribution to the scientific community.

Thank you once again for your thoughtful and constructive feedback. We hope that we have addressed your concerns in the responses provided below.

Comments of Significance:

1. I believe the ROS section could do with refinement of the text. Although it is clear and logical why testing ROS levels was done, how the experiments were done, what was used and why is very unclear. As a result, understanding the figures and context of the data becomes confusing to a non-expert in reactive oxygen species.

Thanks for highlighting that the experiments and results regarding ROS were not clearly explained. We reviewed the text accordingly and added new sentences to make this information clearer to readers. Thank you – much clearer. No further comment on this section.

2. Line 86-87 states: “This finding also indicated that S-phase arrest does not exclusively depend on the presence of the parasite but might be induced by parasite secretory/excretory proteins.” In my opinion, I find this statement weak/not supported by the data and justification for the follow up use of MYR1/HCE1 mutants lacking:

- Authors state that because 64% of the cells at S-phase are infected, then the other 36% arrested cells are not infected because of exported Toxoplasma proteins. However, ~36% of uninfected cells in S-phase, of the total percentage (~35%) of cells in S-phase in infected samples, equals ~12% which could be within statistical variance of the 9.9% of cells that are at in S phase in uninfected samples e.g. the percentage of uninfected cells in S phase might remain stable throughout. Reading between the lines, the authors might be referring to rhoptry protein injection, where tachyzoites inject rhoptry proteins into the host cell but do not completed invasion, leaving a population of uninfected bystander cells with Toxoplasma effector proteins inside (reference: Koshy et al. 2012 Plos Path). However, this is never discussed in the manuscript. Additionally, from the current literature, these proteins are exclusively rhoptry proteins, and there is no data to suggest MYR1 is involved in rhoptry secretion.

We are glad you brought this point to our attention during the revision because we were not aware of how it sounded when isolated from the whole text. First, I apologize for this confusing conclusion regarding our cell cycle results. Secondly, I agree with you that this statement does not fit with our results, and we have erased it from the text. Its much clearer now, thank you. No further comment on this section.

- Although I believe there is justification for following up with KO parasites lines with MYR1 and HCE1 because of my knowledge in the field, this is not well justified in the manuscript. Another sentence or two between lines 87 and 88 stating the reasoning with respect to the literature would go a long way to help the readers understand why the next experiments were taken. The current statement (highlighted above) is in my opinion not enough.

- Similarly, there has been some suggestions that GRA16 was involved in cell cycle and DNA damage (considering P53 modulation), but this was not an effector chosen for study. Justification for this would be helpful, whether in the introduction or in the discussion.

Following your recommendation, we added the information that led us to use the MYR1 and HCE1 mutants in our study. We hope that the results section now provides a better understanding of the results shown and the research pipeline. Please refer to lines 86-96. Thank you – much clearer. No further comment on this section.

3. Figure 2 – no quantitation of the multipolar spindle was provided yet the conclusion that this is changed upon infection is used throughout the manuscript. Additionally, it’s very hard (dear I say almost impossible) for an untrained eye to see what is referred to as the multipolar spindle? There is no marker for the spindle poles other than arrows pointing to an area where not much is visible. I would find better presentative images of this or re-stain and image with a spindle marker rather than b-Catenin. If this cannot be done, I would consider removing or heavily minimising the conclusions around the multipolar spindle poles.

Your observation about what we have presented in Figure 2 is completely right. Since there are a couple of published papers showing that *T. gondii* induces multipolar spindle in several cell types, we allowed

ourselves the freedom to only mention that we also observed this phenomenon only after 3 hpi. Published was shown at 24 hpi only. However, this was only an observation and following your indications, we have erased the image with the chromosome segregation problem as well as our conclusions about this observation. Thank you for being honest about this, and for updating. Just note that I believe a sentence still remains about the multipolar spindles (lines 140-141) and along with a figure reference for a figure that doesn't exist anymore. Please just double check these are deleted. You might also want to remove it from your schematic, this is up to you.

4. The author's data suggests DNA damage happens very early on and in a MYR1-dependent fashion, along with S-phase arrest. However, the authors do not appear to discuss or speculate why the parasite might induce DNA damage? What is the benefit to the pathogen for its survival? Previous theories have suggested that cell cycle arrest could enable Toxoplasma to complete its lytic cycle before the host cell divides. What comes first: does DNA damage cause the cell to arrest in S-phase and the parasite requires this for their survival? OR is DNA damage a consequence of S-phase arrest which is the factor dependent on effector protein export? Although the data presented by the authors cannot fully answer these questions, the questions non-the-less could be included in the discussion for future gaps that could be answered.

We appreciate your insights regarding the correlation between our results and the first stages of infection. We agree that we did not elaborate extensively on our hypothesis linking cell cycle arrest and DNA damage induction. This is because we consider that our results highlight the need to investigate the first minutes of the infection and the potential role of MYR1 in this process. However, the limited scope of our current manuscript makes it not appropriate to hypothesize extensively with only the hint that MYR1 might have a potential role in the host cell cycle arrest and the DNA damage induction.

Our study was initially designed to focus on the first 3h p.i but we were surprised by the earlier effect on cell cycle and DNA damage. Therefore, we decided to leave this topic for further manuscripts, which we are currently working on. Nevertheless, we would take your advice and acknowledge these data as a future research gap in our discussion. We will incorporate some of your comments, as they align with our perspectives. Thank you for your valuable feedback. No problem, I will be interested to see if the field can finally answer why the parasites arrest the host cell cycle. Thank you for updating this. No further comment on this section.

General Comments:

Line 42-45. Has the GRA16 reference, but there is no a reference for the statement that MYR1 indicating it is part of multiprotein complex as suggested in the sentence.

Per your suggestion, we have now included the necessary citations. Thank you for updating this. No further comment on this section.

Add a line showing from where the comet tails were measured in 2A would help understand how the data was produced.

The modification was done in lines 115-123. Thank you for adding in that explanation! No further comment on this section.

It would be helpful to include a sentence in the Figure 5 legend that states vinculin is the loading control.

Thanks for noticing the absence of a description of what was the vinculin's role in the WB. The information was added to the legend. Thank you for updating this. No further comment on this section.

ROS is an abbreviation, please spell out what ROS stands for the first time it is used.

Done. Thank you for updating this. No further comment on this section.

Fig 6A, please clarify in the text/figure legend what WOS is (is it a dye that binds to ROS?) and what it represents in the figure (-ve control for ROS?). Additionally, as 6B is the quantification of 6A, please label them as either 6Ai/6Aii or as one sub figure under 6A. This makes it clearer to the reader that the data is from the same experiment.

Thanks for highlighting the missing information about WOS and the confusion between figures 6A and B. Following your recommendations, we fused 6A and B in only Figure 6A. Also, the definition of WOS (without staining) is now stated in the figure legend and the result text.

Thank you for updating this. No further comment on this section.

The units of ROS measure (absolute count? intensity?) are missing for 6B and C.

We apologize for this confusion; intracellular ROS was measured using the DCFH-DA probe and the values are informed as the mean of fluorescence intensity (MFI) whilst extracellular ROS was detected by using Amplex red staining and the units informed in the graph are relative fluorescence intensity (RFI). Both are written in the graph axes, but we added them now also in the figure legend.

Thank you for updating this. No further comment on this section.

Figure 6C, in the figure legend it would be helpful to include the dye/fluorescence marker used to determine the values.

Done. Thank you for updating this. No further comment on this section.

Is the decrease in intracellular ROS of Tg infected cells dependent on the presence of MYR1? It is known that ROS are often used as an anti-infective defence mechanism in many intracellular pathogens (including Toxoplasma in macrophages), so it is interesting it goes down upon infection here. Maybe MYR1 exported or ROP proteins (which has been shown by Kochanowsky et al. 2020 (mBio) to be the case for Type III strains) help control the levels of ROS for survival.

We appreciate your interest in our results. Initially, we quantified ROS in all three strains using luminometry. This involved incubating cells with luminol and isoluminol. As shown in the graphs below, we did not observe any significant changes in extra- or intracellular ROS levels in MYR1- or HCE1-deficient strains compared to the KU80 strain.

Despite these results indicating no modulation of ROS, previous studies have shown that ROS levels increase in HeLa-infected cells. Therefore, we proceeded with a more sensitive method: FACS analysis of live cells incubated with the DCFH-DA probe. Since no differences were observed across all strains, we focused our analysis on TgRH, as detailed in the manuscript. Thus, the mutant strains were not

analyzed with DCFH-DA by FACS, given that the luminol/isoluminol results suggest that none of the *T. gondii* strains increases intracellular or extracellular ROS.

Regarding your comments on macrophages and other strains, we agree with your hypothesis. We have been working with human and bovine neutrophils as well as macrophages confronted/infected with *Toxoplasma*, but we did not see a positive or negative impact on ROS, even though a recent study described an increase in extracellular ROS in human PMNs (<https://doi.org/10.3389/fimmu.2023.1282278>). However, none of us have tried any mutant strains to assess their effect on innate immunity. We will keep your comments in mind for future experiments to test whether the MYR1 mutant could be important in the immune reaction against infection.

Thanks for the explanation! No further comment on this section.

Figure 6D, NAC+Tg images have barely any invaded tachyzoites in the presentative image. It is hard to evaluate the differences where there are no parasite infected cells.

Thanks for highlighting the differences in the infection shown in Figure 6D. We took new pictures showing similar infections in the field of view and these pictures are in the new version of Figure 6D.

Thank you for updating this. No further comment on this section.

Figure 6D, why do almost all the bystander cells have γ H2Ax expression when they are not infected? This contradicts the authors previous data in figures 1, 3 and 4.

I apologize, but I don't fully understand your question. Therefore, I will provide an explanation based on what I believe you asked. First of all, in Figure 1 there is no quantification about DNA damage foci (γ H2Ax), but maybe this was only a typo problem. Thus, I will refer to only Figures 3 and 4. When we described the DNA damage by the use of γ H2Ax staining, we only showed quantification from the whole n.i. and infected monolayer, therefore, we have no data about the infected cells and bystander cells regarding the DNA damage. Our comparisons are only referred to the non-infected monolayer as an independent experiment and never splitting data from the same infected monolayer (Tg vs bystander or n.i. cells). As we also explained to Reviewer 1, in cell culture conditions, we recognize that cells quickly display DNA damage foci when they experience changes in environmental conditions such as prolonged time outside the incubator, abrupt temperature reductions, or aggressive pipetting during the trypsinization process. Since we work exclusively with primary cells, this can be explained by their unfamiliarity with a 2D structure, where their only contact is with the plastic of the flask and the air above. Also, this could be due to the highly restricted medium that scientists use in the cell culture routine in which, for instance, there is any associated hormonal regulation. Therefore, we have trained ourselves to avoid these perturbations during cell culture. Nevertheless, a baseline percentage of DNA damage in foci-positive cells is always expected. Considering this fact and that our experimental design sought to correlate non-infected (n.i.) and *T. gondii*-infected conditions, we conducted all experimental conditions in all cell donors in parallel.

When counting DNA damage-positive cells, we only included those positive for γ H2AX, normalized by the number of DAPI-positive cells in the same field of view. Consequently, figure 4B only shows those cells, in the monolayer, that were positive for DNA damage. What was surprising is that G2-phase cells showed higher DNA damage in non-infected conditions. However, we do not have a definitive explanation for this. One possibility is that G2-phase cells, having just completed DNA synthesis, maybe in the process of correcting DNA damage through intracellular repair pathways. Studies on cell cycle control have shown that DNA damage checkpoints in the G1 and G2 phases work to repair any damage produced in previous stages, sometimes arresting the cell to allow time for repair (DOI:<https://doi.org/10.1016/j.cels.2017.09.015>).

Thank you for explaining! No further comment on this section.

Statistical comments:

Figures 2, 3 and 5 show the individual samples points on the bar graphs in addition to the error bars. This is very good practise and helps the reader critical analyse the data and statistical output. Please also do this for Figures 1, 4, and 6.

All graphs were amended with bars and points. The only exception was Figure 1B where adding the p-value makes the graph hard to interpret. However, the value of p was added to the Figure legend.

Thank you for amending the graphs. I believe 4B has asterisks and no value of the stats in the figure legends? Please update this one as well.

I noticed all the data is tested using non-parametric tests, can the authors please include in methodology whether they did a test for non-normalcy prior to this decision and what this test was.

In the methodology the authors state the data shown is in mean +/- SD, however, in figure 1 legend its states median +/- SD. Please clarify which is shown. I am not an expert in statistics, however from my understanding the statistical tests used are rank based, which should suggest the median would be represented. If the median is represented, I believe SD is not the accurate variance to show, as SD is based on normal (e.g. mean) data. I would discuss the most accurate way to present this with a more experienced statistician. The rest of the figure legends say mean +/- SD, but once again, I question if this is the most accurate way to represent the data if non-parametric tests were used.

Thanks for this comment because it was something that we have found several times in our research statistical analysis. The chosen tests were based on the assumptions of the tests and the distribution of the data. Regarding the results calculated as % of cells with X effect, since the data is already normalized (%) and thus continuous, the conventional recommendation is to run non-parametric tests. In this case, the selection of the test was not based on the result of a normality test but on the type of data (i.e. percentages). In the case of discrete data (i.e. long of the tails), some datasets did not pass the normality test, based on the results of D'Agostino & Pearson and Shapiro-Wilk normality tests. This information is now added in the material and Methods section.

Thank you for updating this. No further comment on this section.

We completely agree with you that the median is the correct statistical parameter to be reported on the plots when non-parametric tests are used, as correctly pointed out by the reviewer since are based on median comparisons and then ranks. However, I think the readers (and sometimes reviewers, editors etc) are more used to assuming that the mean \pm SD is represented in the figures, even though the median is more appropriate. Hopefully, the reviewer agrees to maintain mean \pm SD as the way to show the graphs, just because is more conventional.

Unfortunately, I disagree. You cannot show mean \pm SD when the test you did was a non-parametric test. This is miss-leading. How we do our statistics should be accurate and presentative of the correct test/statistical analysis, not what others assume it is or what is conventional. This is why we write what we have done in the methods and figure legends. Please amend to show median with QR- a box and whiskers plot with individual data points overlayed is a good place to start, or a violin plot.

In some of the figures, stars are shown to represent statistical significance, however, no key is provided in the legends about what the stars mean. Other times the p values are given directly on the figure. Authors should pick one for the entirety of the manuscript. I personally prefer the p-values are given directly on the figures.

Thanks for noticing this discrepancy. Following your recommendation we changed all asterisks by p-value in graphs to show directly whether they were or not statistically significant. The only exception

is Figure 1B in which we kept the asterisks to avoid an overcrowded graph, but we informed the p-value of each statistic written in the graph.

Thank you for amending the graphs. I believe 4B has asterisks and no value of the stats in the figure legends?

Authors should double check the CDK2 WB p-value is correct. At $p=0.053$ it would technically be non-significant, other for the sample distributions shown in the figure that value seems high. If 0.053 is correct, the authors should consider rewording their language and conclusions regarding ATM vs ATR.

You are correct about CDK2 statistical significance. We are aware that they were non-significant, and we kept it only because we wanted to show that even though CDK2 expression was evident lower, they were not statistically significant. However, we noticed with your question that it leads to misunderstanding, and we erased the p-value.

Thank you for updating this. No further comment on this section.

Figure 6 legend states a “non-parametric t-test” was used, please change this to correctly state which test was used by name like the other legends.

All figure's legends were changed indicating the right statistical test used.

Thank you for updating this. No further comment on this section.